



# Modelling spatiotemporal variations of the canopy layer urban heat island in Beijing at the neighbourhood-scale

Michael Biggart[1], Jenny Stocker[2], Ruth M. Doherty[1], Oliver Wild[3], David Carruthers[2], Sue Grimmond[4], Yiqun Han[5,6], Pingqing Fu[7,8], Simone Kotthaus[4,9]

[1]School of Geosciences, The University of Edinburgh, Edinburgh, UK

[2]Cambridge Environmental Research Consultants, Cambridge, UK

[3]Lancaster Environment Centre, Lancaster University, Lancaster, UK

[4]Department of Meteorology, University of Reading, Reading, UK

[5]State Key Joint Laboratory for Environmental Simulation and Pollution Control, College of Environment Sciences and Engineering, Peking University, Beijing, China

[6]Environmental Research Group, MRC Centre for Environment and Health, King's College London, London, UK

[7]Institute of Atmospheric Physics, Chinese Academy of Sciences, Beijing, China

[8]Institute of Surface-Earth System Science, Tianjin University, Tianjin, China

[9]Institut Pierre Simon Laplace, École Polytechnique, Palaiseau, France

*Correspondence to*: Michael Biggart (michael.biggart@ed.ac.uk)

**Abstract.** Information on the spatiotemporal characteristics of Beijing's urban-rural near-surface air temperature difference, known as the canopy layer urban heat island (UHI), is important for future urban climate management strategies. This paper investigates the variation of near-surface air temperatures within Beijing at a neighbourhood-scale resolution (~ 100 m) during winter 2016 and summer 2017. We perform simulations using the urban climate component of the ADMS-Urban model with land surface parameters derived from both Local Climate Zone classifications and OpenStreetMap land use information. Through sensitivity simulations, the relative impacts of surface properties and anthropogenic heat emissions on the temporal variation of Beijing's UHI are quantified. Measured UHI intensities between central Beijing (Institute of Atmospheric Physics) and a rural site (Pinggu) during the Atmospheric Pollution and Human Health in a Chinese Megacity (APHH-China) campaigns, peak during the evening at ~ 4.5 °C in both seasons. In winter, the nocturnal UHI is dominated by anthropogenic heat emissions but is underestimated by the model. Higher resolution anthropogenic heat emissions may capture the effects of local sources (e.g. residential buildings and adjacent major roads). In summer, evening UHI intensities are underestimated, especially during heatwaves. The inability to fully replicate the prolonged release of heat stored in the urban fabric may explain this. Observed negative daytime UHI intensities in summer are more successfully captured when surface moisture levels in central Beijing are increased. However, the spatial correlation between simulated air temperatures and satellite-derived land surface temperatures is stronger with a lower urban moisture scenario. This result suggests that near-surface air temperatures at the urban meteorological site are likely influenced by fine-scale green spaces that are unresolved by the available land cover data and demonstrates the expected differences between surface and air temperatures related to canopy layer advection. This study lays the foundations for future studies of heat-related health risks and UHI mitigation strategies across Beijing and other megacities.

## 1   Introduction

The urban heat island (UHI) phenomenon describes the positive temperature difference between urban environments and their surrounding rural areas (Oke, 1982; Arnfield, 2003; Grimmond et al. 2010). Within the urban atmosphere, distinct UHIs can be defined for the urban canopy layer, extending from the ground surface to mean building height, and the urban boundary layer, covering the remainder of the mixing layer above the urban canopy (Voogt and Oke, 2003). Different mechanisms drive



the development of each atmospheric UHI (Oke, 1982). Here, we focus on the canopy layer UHI, with raised temperatures resulting from the morphology of urban structures and street canyons, trapping incoming shortwave (SW) and outgoing longwave (LW) radiation, and the replacement of natural, permeable surfaces with impervious materials, such as concrete, which alters the urban surface heat energy balance (Estoque et al. 2017; Ao et al. 2018). The removal of vegetation and increased runoff of surface water lowers the proportion of net radiation partitioned to latent heat flux thereby reducing daytime urban evaporative cooling (Li et al. 2015; Wang et al. 2017; He et al. 2018). Heat stored throughout the day within the high thermal admittance urban fabric is released into a stabilising boundary layer at night creating a strong nocturnal UHI effect (Anandakumar, 1999; Grimmond and Oke, 1999). The continuous emission of heat from anthropogenic activities further enhances the urban-rural temperature contrast (Sailor, 2011; Gabey et al. 2019).

In China, urbanisation has occurred rapidly in recent decades, with 59.6 % of the population reported to be living in urban areas in 2018, compared to 19.4 % in 1980 (National Bureau of Statistics, 2018; The World Bank, 2020). The UHI effect is well known to exacerbate and prolong extreme temperature events (Li et al. 2015; Jiang et al. 2019). As heatwaves are becoming more frequent in our warming climate (Krayenhoff et al. 2018; Zhao et al. 2018), the number of people in China left vulnerable to heat-related illnesses is increasing (Tan et al. 2010; Bai et al. 2014; Gu et al. 2016). Most at risk from illnesses such as heat stroke are the elderly (Gu et al. 2016), of particular concern in China owing to its aging population (Li et al. 2016), and those without air conditioning (Zhao et al. 2018). The latter is of most significance at night when residents are at home and the UHI intensity (UHII), the magnitude of the urban-rural near-surface air temperature difference, is strongest (Liu et al. 2007; Wang et al. 2017; He et al. 2020). Chen et al. (2016) estimates that residents of Chinese megacities may spend up to 40 % more time under extreme heat stress compared to those living in adjacent rural areas.

A comprehensive understanding of the relative importance of surface radiative properties, urban morphology and anthropogenic heat emissions (AHEs) in driving spatiotemporal UHI variations across cities is essential for the development of successful urban planning strategies aimed at reducing the heat-related health burden. A common approach to quantifying a city's UHI is to look at differences between near-surface air temperatures measured at in-situ urban and rural meteorological stations (Liu et al. 2007; Wang et al. 2017; Jiang et al. 2019). However, sparse observations (Santamouris, 2015) cause sharp temperature gradients between distinct urban microclimates to be unresolved (Hamilton et al. 2014; Aktas et al. 2017). Furthermore, variations in building morphology and other surface properties, combined with the lack of knowledge of the urban canopy observational footprints constrain UHI comparisons between cities (Oke, 2004; Schatz and Kucharik, 2014). A comparison of UHIIs observed across multiple Chinese cities by Jiang et al. (2019) found that Shanghai's daily maximum UHII could alternate between afternoon and evening hours, depending on the chosen rural reference site, due to its coastal proximity and sea breeze effects.

Satellite-derived land surface temperatures (LST) allow land cover and temperature to be studied across cities lacking dense near-surface air temperature observations (Kato and Yamaguchi, 2005; Zhou et al. 2013; Estoque et al. 2017). However, comparisons between the spatiotemporal variability of LSTs and near-surface air temperatures are limited by the differing controls of both variables, notably the relative importance of advection to the surface energy balance partitioning (Chandler, 1965; Roth et al. 1989). Lack of information on surface radiative properties, building geometry and urban atmospheric composition can further limit the use of LSTs by making the surface temperature difficult to derive (Morrison et al. 2020). Specifically, radiance measurements made by remote thermal sensors need to be corrected for spatially varying urban surface emissivities; walls or roofs of buildings may become oversampled depending on the satellite viewing angle; and cloud cover and urban pollution can strongly attenuate the upwelling thermal radiance, restricting the selection of satellite images to clear days to minimise signal interference (Roth et al. 1989; Voogt and Oke, 1998; Voogt and Oke, 2003; T. Wang et al. 2019).

Urban climate modelling is frequently undertaken to enable the testing of UHI mitigation strategies and the investigation of critical issues such as how the UHI impacts heat wave events and pollution dispersion (Wang et al. 2013; Chen et al. 2016; Fallman et al. 2016). Regional-scale climate models such as the Weather Research and Forecasting model (WRF)





coupled with urban canopy modules that simulate the impact of urbanisation on climate at citywide scales are commonly used for these analyses (Loridan et al. 2010; Wang et al. 2013). Higher-resolution local-scale (~1 km) UHI effects can also be

explored in detail with urban energy balance models, such as the Surface Urban Energy and Water Balance (SUEWS) model (Alexander et al. 2015), which incorporates detailed land cover data to calculate local perturbations to surface heat fluxes (Grimmond et al. 2010). However, the availability of land cover information in developing countries, where heat-related mortality rates in cities are highest (Kjellstrom et al. 2009), often restricts such studies. In fact, the Intergovernmental Panel on Climate Change's (IPCC) fifth assessment report specifically highlighted the lack of detailed global urban land use datasets

(IPCC, 2014). Stewart and Oke's (2012) thermal classification scheme for neighbourhoods, termed local climate zones (LCZ), provides one solution. It has 10 urban and 7 rural classes that can be mapped by a variety of methods including using remote sensing data combined with local expert-based knowledge (Bechtel et al. 2015). LCZs are distinguished based on surface cover, structure, material and human activity, as they are designed to standardise the characterisation of near-surface temperature measurement sites in both urban and rural locations (Stewart and Oke, 2012; Ching et al. 2018).

Here we investigate the key processes driving neighbourhood-scale (~100m) near-surface air temperature variations across Beijing using the ADMS-Urban Temperature and Humidity model (hereafter ADMS-Urban). This local-scale urban climate model has been used for studies in megacities of contrasting climates such as London (Hamilton et al. 2014; Aktas et al. 2017) and Kuala Lumpur (K. Wang et al. 2019). As our simulations cover the two field campaign periods of the Atmospheric Pollution and Human Health in a Chinese megacity (APHH-China) programme (Shi et al. 2019), the results may assist the

interpretations of related air quality measurement and modelling studies (Shi et al. 2019; Biggart et al. 2020; Squires et al. 2020; Zhao et al. 2020). ADMS-Urban studies in London focused only on specific urban developments (Hamilton et al. 2014) and building materials (Aktas et al. 2017), whereas in Kuala Lumpur the effects of AHEs were excluded (K. Wang et al. 2019). By adding the locations of buildings, green spaces and waterways using OpenStreetMap (OSM), we build on the previous implementation of LCZs for UHI simulations in Dublin (Alexander et al. 2015).

Detailed descriptions of the ADMS-Urban model, its meteorological inputs and the development of surface parameter and AHE datasets are provided in Sect. 2. In Sect. 3, simulated neighbourhood-scale spatiotemporal near-surface air temperature variations across urban Beijing are evaluated using in-situ near-surface air temperature measurements and satellite-derived LSTs, with the impacts of extreme temperature events on Beijing's UHI also explored. Section 4 provides a summary of this work's findings, along with suggestions for study improvement and future applications.

## 2 Methodology

### 2.1 ADMS-Urban model description

The neighbourhood-scale ADMS-Urban climate model calculates local perturbations to upwind vertical profiles of temperature and humidity in response to spatially varying neighbourhood-scale surface parameters. Calculations of upwind profiles, based on rural near-surface meteorological measurements, depend on the planetary boundary layer height (PBLH), surface roughness

length ($z_0$) and the stability parameter PBLH/$L_{MO}$, where $L_{MO}$ is the Monin-Obukhov length, a measure of the relative importance of mechanical turbulence and buoyancy (Hood et al. 2018; CERC, 2020). Calculation of the local perturbations to these vertical profiles arises from surface thermal and morphological parameters modifying heat and moisture processes, allowing a 3-D temperature and humidity field to be modelled across the domain.

    The temperature and humidity perturbations are calculated by solving a coupled system of equations that govern heat

and moisture processes at the ground (Carruthers and Weng, 1992; Raupach et al. 1992). The conservation of heat's boundary conditions are dependent on moisture via the latent heat flux; and evaporative processes are temperature dependent. The equations account for the spatial variation of surface heat fluxes, moisture, mean airflow and vertical turbulent diffusion (CERC, 2018). Heat fluxes are governed by variations in net radiation ($Q^*$) and storage heat; airflow and turbulent diffusion





are controlled by variations in surface roughness ($z_0$); and the moisture terms depend on the surface's resistance to evaporation

which accounts for both surface wetness and sub-surface moisture, with water surfaces having near-zero surface resistance to

evaporation (Table 1).

Upwind $Q^*$ is determined from near-surface, hourly meteorological measurements representative of conditions

upwind of the model domain, following Eq. (1) (Holstag and van Ulden, 1983):

$$Q^* = \frac{(1-r)K^+ - 5.67 \times 10^{-8} \, T^4 + 5.31 \times 10^{-13} \, T^6 + 60\left(\frac{C_L}{8}\right)}{1.12} \tag{1}$$

where the first term represents the incoming SW radiation ($K^+$) absorbed by the surface, which is dependent on the upwind

surface albedo ($r$). The second term in Eq. (1) accounts for the LW radiation released by the surface, a function of the upwind

near-surface air temperature, T, and the downwelling LW radiation from the gaseous atmosphere and cloud cover ($C_L$) are

given by the third and fourth terms, respectively (Holstag and van Ulden, 1983).

The upwind storage heat flux ($G$) is calculated as a function of $Q^*$ according to Eq. (2) (Camuffo and Bernardi, 1982):

$$G = a_1 \frac{dQ^*}{dt} + a_2 Q^* + a_3 \tag{2}$$

where  $a_1$(h), $a_2$, and $a_3$ (W m$^{-2}$) are coefficients that depend on the land cover and surface moisture (Grimmond et al. 1991;

Grimmond and Oke, 1999), and vary sinusoidally throughout the year (Keogh et al. 2012; Sun et al. 2017). For this application,

the coefficients selected correspond to a hybrid of urban and rural surface properties (Grimmond and Oke, 1999) that allow

for both the influence of urban materials on heat exchanges as well as the advection of nighttime stable atmospheric conditions

from the rural to urban areas, with annual mean $a_1$=0.7 h, $a_2$=0.3 and $a_3$=-7.5 W m$^{-2}$ (CERC, 2018). These values respectively

represent the high proportion of $Q^*$ absorbed into the ground during the day in urban areas and re-released overnight, asymmetry

relating to the urban fabric absorbing more heat in the early part of the day, and the model requirement that the upwind PBLH

corresponds to stable conditions at night. In reality, in urban areas, the nocturnal release of stored heat is often sufficient to

produce an upward convective heat flux and hence maintain an unstable PBL (Oke et al. 1999); in ADMS-Urban, this is

modelled via perturbations to the upwind conditions. The model accounts for the diurnal variation of ground heat storage and

release by summing the daytime storage heat flux and releasing it linearly throughout the evening.

The magnitude of $Q^* - G$ determines the amount of energy available for partitioning between the latent ($LE$) and

sensible heat fluxes ($H$). Local perturbations ($\Delta$) to the upwind surface heat fluxes are calculated based on relative differences

between upwind and local surface parameters, according to the surface energy balance equation:

$$\Delta Q^* - \Delta G - \Delta LE = \Delta H \tag{3}$$

The model requires the following surface parameters (Table 1): thermal admittance (J K$^{-1}$ m$^{-2}$ s$^{-1/2}$), surface resistance to

evaporation (s m$^{-1}$), albedo, surface roughness length ($z_0$) (m) and normalised building volume (NBV) (m). Local $\Delta Q^*$ values

depend on the urban-upwind differences in albedo and NBV. $\Delta G$ is based on spatial differences in the surface's thermal

properties (thermal admittance). Perturbations to energy lost through evaporation at the surface ($\Delta LE$) depend on locally

varying surface resistance to evaporation, air temperature and surface moisture content. $\Delta H$ represents spatially varying rates

at which heat energy is transferred from the surface to the atmosphere by convection. Local perturbations to the upwind

temperature profile due to surface characteristics depend on the cumulative effect of $\Delta H$ and the local surface roughness length

($z_0$) accounting for the impact of building morphology on advection and turbulent diffusion and hence heat transfer processes.

### 2.2 Study period and meteorological input data

Simulations are performed for the winter (10 November 2016 – 10 December 2016) and summer (17 May 2017 – 21 June

2017) APHH-China measurement campaigns (Shi et al. 2019). These are periods of interest given the expected impact of the





urban canopy layer UHI on air pollution dispersion. Rural meteorological data for driving the model are available for the duration of both campaigns.

        Upwind meteorological conditions are defined for this study using a single set of meteorological measurements. Near-surface air temperature and relative humidity (~ 8 m above ground level and ~ 50 m above sea level) recorded at the rural APHH-China field campaign site (Han, 2019) in Xibaidian village, Pinggu (40.17 °N, 117.05 °E) (hereafter the rural site), are

used along with wind speed, wind direction and cloud cover data from the Beijing Capital International Airport Meteorology Observatory. Pinggu is located ~ 60 km north-east of the urban APHH-China field site at IAP (39.97 °N, 116.37 °E) (hereafter the urban site). The World Meteorological Organization (WMO) Beijing Capital International Airport Meteorology Observatory (https://ncdc.noaa.gov/, last access: 19 May 2020) is situated approximately halfway between the two sites, within the Sixth Ring Road (Fig. 1). Measurements of wind speed and direction from the airport site are used as the 'rural' data as it

is subject to WMO quality assurance and is likely to be less influenced by local frictional effects of neighbouring small buildings and vegetation due to its more exposed location. During the winter period, the predominant wind direction was from the north-east, situating the model domain downwind of the rural site. During the summer period, winds from the north and east are the most prevalent, however a growing frequency of south-westerly winds is also observed. Therefore, summer air temperatures measured at the rural site may be influenced at times by heat advection from urban Beijing.

At the urban site, a ceilometer gathered attenuated backscatter that were analysed with the CABAM algorithm (Kotthaus and Grimmond, 2018) to provide the mixing layer height, the atmosphere's lowest layer in contact with earth's surface resulting from turbulent exchange (Shi et al. 2019; Hertwig et al. 2020). This was assumed to equate to the PBLH.

### 2.3 ADMS-Urban surface parameters

The model domain is the area contained within Beijing's Sixth Ring Road (Fig. 1). Thermal and morphological properties

covering the domain are derived from OpenStreetMap (OSM) data (https://openstreetmap.org/, last access: 9 March 2020) and LCZs mapped by the World Urban Database and Access Portal Tools (WUDAPT) project (http://www.wudapt.org/cities/in-asia/, last access: 19 May 2020). The function and plan area of specific buildings, green spaces and waterways are obtained from OSM (Fig. 2). Table A1 provides a full list of the OSM land use types and the thermal admittance, surface resistance to evaporation and albedo values assigned to each, based on data reported in the literature (Oke, 1982; Cox et al. 1999; Hamilton

et al. 2014; CERC, 2018; K. Wang et al. 2019).

        OSM land use information is overlaid onto Beijing's LCZ classes, which are mapped at 100 m resolution (Fig. 2). The methodology for generating LCZ maps for specific cities is well-documented (Stewart and Oke, 2012; Bechtel et al. 2015; Ching et al. 2018). For Beijing, there are 9 urban and 6 rural LCZ types (Fig. 2; Table 1). The surface parameters assigned to each LCZ (Table 1) are based on ranges of values suggested by Stewart and Oke (2012), derived from previous urban climate

observational and numerical modelling studies. Following Ching et al. (2018), thermal admittance values are assigned by selecting the midpoints of the Stewart and Oke (2012) ranges. Estimates of surface resistance to evaporation are not provided with the LCZ data, therefore we use literature-reported values (Oke, 1982; Cox et al. 1999; Hamilton et al. 2014; CERC, 2018). To ensure we capture sufficient heterogeneity across central urban areas, a higher surface resistance to evaporation of 200 s m$^{-1}$ is given to urban LCZs described by Stewart and Oke (2012) as consisting entirely of impervious materials (LCZ 1-3, 8

and 10), with 150 s m$^{-1}$ assigned to those comprising abundant pervious land cover (e.g plants and trees) (LCZ 4-6 and 9). The albedo values selected for each LCZ are the lower bounds of Stewart and Oke's (2012) ranges, closely matching the literature-reported albedo values used for the OSM data. Surface roughness lengths ($z_0$) correspond with the Davenport classes (Davenport et al. 2020) assigned to each LCZ by Stewart and Oke (2012). The NBV are calculated as the product of the midpoints of Stewart and Oke's (2012) ranges for roughness element height and building surface fraction.

A spatially weighted mean of the surface parameters is calculated at 100 m resolution (Fig. 3), matching the resolutions of the LCZ data and the model calculation grid (~ 105 m). A 1 km-width border around the edge of each map in





Fig. 3 defines the upwind conditions. Differences between upwind surface parameters, representing conditions at the rural site, and those within the model domain are used to calculate the urban temperature perturbations (Sect. 2.1). Upwind parameters (Table 1) were derived from Stewart and Oke's (2012) ranges for LCZ 9 based on its definition as a 'natural setting with sparsely arranged buildings and abundant pervious land cover', closely matching the environment at the rural site in Pinggu (Shi et al. 2019). Stewart and Oke's (2012) LCZ 9 parameter values were modified to those given in Table 1 following model sensitivity tests, with thermal admittance most notably reduced to 600 $J\,K^{-1}\,m^{-2}\,s^{-1/2}$, accounting for the large increment of heat stored in Beijing's urban fabric relative to neighbouring rural areas. Altitude effects on temperature are accounted for with use of terrain elevation data (Fig. 3f) (https://www.usgs.gov/land-resources/eros/coastal-changes-and-impacts/gmted2010, last access: 1 April 2020). Urban Beijing is situated on a plain at an altitude of ~ 50 m, but mountainous terrain across rural areas lies to the north and west of the city (Fig. 3f).

**2.4 Anthropogenic heat emissions (AHE)**

The anthropogenic heat flux is not included in Eq. (3) and needs to be accounted for. The main sources include power, industry, transportation, residential and commercial building use, and human metabolism (Sailor, 2011; Lu et al. 2016; Yu et al. 2018). Our values are based on a mean summer AHE value estimated by Dou et al. (2019) at IAP using the Large scale Urban Consumption of Energy model (LUCY) (Lindberg et al. 2013). Their daily mean aggregate summer AHE (48.1 $W\,m^{-2}$) is derived from national- and provincial-level energy consumption data and spatial proxy information (population density, gross domestic product and vehicle ownership rates) (Dou et al. 2019), a methodology also applied to other worldwide megacities such as London (Gabey et al. 2019) and Shanghai (Ao et al. 2018). For this study, we generate a 3 km resolution grid (Fig. 4) of aggregate AHEs, representing the sum of all AHE source sectors, by scaling the IAP value (situated in LCZ 1) to the other LCZ classes based on the relative values given by Stewart and Oke (2012) for each LCZ classification.

Single diurnal and monthly profiles are applied to the AHEs based on the profiles estimated by Lu et al. (2016). Lu et al. (2016) derived annual sectoral AHE totals for all provinces and municipalities in China from the officially published energy consumption statistics, obtained from a variety of sources (e.g. China Energy Statistical Yearbook). Weighted means of the diurnal and monthly temporal profiles for each sector, from Lu et al. (2016), are determined in relative proportion to the total annual sectoral AHEs estimated for the Beijing municipality. The nocturnal contribution from transportation is increased, following Biggart et al. (2020), to account for the influx of heavy-duty diesel trucks (HDDT) into urban Beijing after the daytime ban within the Fourth Ring Road (Zhang et al. 2019).

The resulting monthly and diurnal aggregate AHE profiles (Fig. 5) have winter and summer peaks primarily caused by elevated heating and cooling activities, respectively, in residential and commercial buildings (Ichinose et al. 1999). Emissions are assumed to be proportional to electricity and heating fuel consumption (Lu et al. 2016). Elevated daytime AHEs with late morning and early evening peaks are predominantly from the combination of daytime commercial building thermal regulation and residential cooking and heating, with a smaller contribution from rush hour-related traffic volume maxima (Lee et al. 2009; Ao et al. 2018).

AHEs are modelled as individual plumes with the ADMS-Urban air quality model (Owen et al. 2000; CERC, 2018) and build upon the local temperature perturbations caused by spatially varying surface characteristics (Sect. 2.1). All AHEs are assumed to be released into the atmosphere, then dispersed assuming quasi-Gaussian distributions driven by the upwind meteorological variables, PBLH and the PBLH/$L_{MO}$ stability parameter.

**2.5 Model evaluation**

The magnitude and temporal variability of Beijing's winter and summer canopy layer UHI are evaluated using hourly near-surface air temperature measurements. Unlike city-scale urban climate modelling studies which commonly compare regional-scale climate model output with the nearest meteorological measurement station to a model grid box (Chen et al. 2016;





Fallmann et al. 2016; Chen et al. 2018), ADMS-Urban can simulate air temperatures at specific locations corresponding to the exact coordinates of measurement sites. We compare UHIIs modelled at the urban site at IAP, for the winter and summer

periods, with measured UHIIs, calculated as the difference between near-surface air temperatures observed at the urban and rural sites. Air temperatures at the urban site are recorded at ~8 m above ground level and ~ 50 m above sea level, matching the measurement height and terrain elevation of the temperature sensor at the rural site (Sect. 2.2). The following statistical performance measures are used for model evaluation:

$$root\ mean\ square\ error\ (RMSE) = \sqrt{\frac{1}{n}\sum_{i=1}^{n}(O_i - M_i)^2} \qquad (4)$$

$$fractional\ bias\ (Fb) = \frac{\overline{M} - \overline{O}}{0.5(\overline{O} + \overline{M})} \qquad (5)$$

$$Pearson's\ correlation\ coefficient\ (R) = \frac{1}{n-1}\sum_{i=1}^{n}\left(\frac{M_i - \overline{M}}{\sigma_M}\right)\left(\frac{O_i - \overline{O}}{\sigma_O}\right) \qquad (6)$$

where $n$ denotes the total number of matching hourly modelled ($M$) and observed ($O$) UHIIs; $\overline{M}$ and $\overline{O}$ indicate mean modelled and observed UHIIs, respectively, and $\sigma$ is the standard deviation.

Further measured-modelled air temperature comparisons are not possible due to a lack of access to other near-surface air temperature observations. However, satellite-derived LSTs are readily available for the Beijing area and, although the ADMS-Urban model does not calculate LST, the correlation between neighbourhood-scale spatial variations of modelled near-

surface air temperature and satellite-derived LST provides a useful comparison for model evaluation. Landsat 8 Operational Land Imager (OLI)/Thermal Infrared Sensor (TIRS) thermal band TIRS-1 data (https://www.usgs.gov/land-resources/nli/landsat, last access: 1 April 2020) is acquired at 100 m resolution, matching the modelled spatial resolution. LSTs are retrieved using the single-channel algorithm developed by Jiménez-Muñoz et al. (2009). We focus on the summer for this analysis given the stronger incoming SW, compared to winter, and hence the expected more direct relationship between LSTs

and near-surface air temperatures (K. Wang et al. 2019), with near-surface air temperatures strongly influenced by the upwards flux of heat from the surface. Landsat 8 LSTs are available for 23 May 2017 at 10:53 am (local time), a time with minimal cloud cover, and are compared with near-surface air temperatures simulated on the same day at 11:00 am, both sampled on the same 100 m resolution grid. Due to the difficulty in determining the extent to which AHEs impact LSTs, reported in previous studies (Kato and Yamaguchi, 2005; Wang et al. 2017), LSTs are compared with simulations that exclude AHEs.

**2.5 Model experiments**

We perform simulations with four distinct model configurations, each involving different surface property or AHE scenarios (Table 2). The experiments are designed to identify how the modelled UHII can be optimised to best represent the observations by use of appropriate input parameters, thus providing insight into the key processes driving Beijing's UHI in winter and summer. The Base case uses the initial surface parameters and AHEs (Sect. 2.3 and 2.4). Gabey et al. (2019) showed that the

top-down approach for estimating AHEs, as used for the AHEs adopted for this study, tends to underestimate hotspots associated with dense inner-city road networks and compact building developments. Therefore, Base case AHE values are increased by 50 % in the AHE50 case (Table 2). Given the unexpectedly high latent heat flux values measured previously at the IAP site (Dou et al. 2019), the surface resistance to evaporation values are reduced to 150 s m$^{-1}$ in LCZ categories 1-3, 8





and 10 (Table 1), and for all urban OSM land use types (Table A1) for the Evp150 case (Table 2). This matches the surface resistance to evaporation assigned to the upwind domain. The noAHE case is designed to isolate the contribution to Beijing's UHI from land surface characteristics alone, hence AHEs are excluded.

### 3 Results and Discussion

First, we evaluate UHIIs simulated at the urban site for the winter and summer periods (Sect. 3.1). The neighbourhood-scale spatial variations of modelled near-surface air temperatures are assessed using satellite-derived LSTs in Sect. 3.2. Section 3.3
investigates the extent to which spatial temperature patterns vary throughout the day, focussing on the spatiotemporal characteristics of neighbouring, but distinctly different, urban microclimates. Summer heat wave events are identified in Sect. 3.4 and their impact on Beijing's UHI is discussed.

### 3.1 Model evaluation of diurnal UHII in winter and summer

The winter and summer period mean observed canopy layer UHIIs (IAP-Pinggu) are 3.1 °C and 1.8 °C, respectively. These
values are similar to those derived previously from a dense network of ground measurements in Beijing (Yang et al. 2013; Wang et al. 2017). Yang et al. (2013) reported winter and summer averages across central Beijing (within the Fourth Ring Road) of 2.4 °C and 1.5 °C, respectively. Base case simulated UHIIs are underestimated for both periods compared to the measurements, with the winter values less than half the observed UHIIs (RMSE = 2.90 and Fb = -0.76; Table 3). The summer underestimation is lower, however the RMSE is proportionally higher (Table 3).

The UHI is known to have distinct diurnal characteristics (Oke et al. 2017), hence analysis of the model's ability to simulate the diurnal UHII variation can inform on the daily varying contributions to urban climate perturbations from surface energy balance changes, AHEs and background meteorology. Measured UHIIs in winter and summer peak during the evening at ~ 4.5 °C (Fig. 6). However, other studies have found Beijing's canopy layer UHII maximum to be up to a factor of 2 greater in winter compared to summer months (Liu et al. 2007; Wang et al. 2017). They attributed this to stronger winter AHEs from
elevated energy consumption for building heating systems (Lu et al. 2016), which readily accumulate in Beijing's frequently stable and shallow winter nocturnal boundary layer (Zhang et al. 2016), and more efficient nocturnal cooling in rural areas under frequent strong winter temperature inversions (Yang et al. 2013). Summertime nocturnal UHIIs are influenced more by the delayed release of heat stored within the urban fabric throughout the day (Oke et al. 1999; Wang et al. 2013), with a smaller proportion attributed to AHEs from residential air conditioning units (Zhao et al. 2018).

Simulated evening UHII maxima are underestimated compared to measurements for all scenarios, with Base case values ~ 2.5 °C lower than measurements from 19:00 to 06:00 in winter and ~ 2 °C lower between 21:00 and 04:00 in summer (Fig. 6). However, previous studies of Beijing's evening canopy layer UHI in summer measured at different central urban meteorological sites have reported mean values between 2 and 3 °C (Yang et al. 2013; Wang et al. 2017; Jiang et al. 2019), agreeing closely with our simulated results (Table 4; Fig. 6b). This suggests that Beijing's canopy layer UHI exhibits strong
spatial variability and that our local site characteristics at IAP may be incorrect.

Increasing AHEs by 50 % reduces nocturnal measured-modelled discrepancies in winter by ~ 0.7 °C (Table 4; Fig. 6a). However, a substantial model underestimation persists (~ 2 °C), with nighttime RMSE and fractional bias values of 3.11 °C and -0.58, respectively (Table 4). These low simulated nocturnal UHIIs are likely related to the use of coarse resolution (3 km) aggregate AHEs that fail to resolve strong, local emission sources around the urban site at IAP. The IAP temperature
sensor is located ~ 110 m south of a busy road (Biggart et al. 2020) and is surrounded by high-rise buildings (Fig. 1), both of which are likely strong sources of anthropogenic heat, with nearby buildings also hindering the dispersion of heat at night. Additionally, Cao et al. (2016) found a strong correlation between high concentrations of particulate matter over urban areas and the nocturnal UHI for several megacities in China due to enhanced atmospheric absorption and re-emission of LW





radiation back to the surface, reducing the evening cooling rate. The ADMS-Urban radiation model does not account for
nocturnal warming effects from haze pollution, which has been shown to change surface energy and water partitioning in
Beijing (Kokkonen et al. 2019).

Summer UHII underestimations at night, relative to measurements, are similarly reduced by ~ 0.6 °C with increased
AHEs (Table 4; Fig. 6b), reflected by a fractional bias improvement from -0.61 to -0.4 (Table 4). The remaining measured-
modelled discrepancy is likely related to the model's determination of nocturnal ground heat flux, with the restriction that
modelled upwind PBL conditions remain stable at night (CERC, 2018). However, the evening release of stored heat in other
dry, densely built megacities in summer, such as Mexico City (Oke et al. 1999), has been found to be of sufficient magnitude
to maintain an upward flux of convective heat throughout the evening, therefore prolonging the warming of near-surface air.

The lowest measured urban-rural near-surface air temperature differential occurs between 13:00 and 14:00 in both
the winter and summer periods (Fig. 6). In summer, the observed mean UHII minimum is negative (-1.1 °C). Daytime UHIIs
are largely controlled by the balance between (a) the urban-rural evapotranspiration differences based on vegetation amounts
and irrigation behaviour in both areas (Oke et al. 1982; Grimmond et al. 1993; Estoque et al. 2017; He et al. 2020), and (b) the
large storage heat fluxes associated with the building volumes and impervious materials of high thermal conductivity and heat
capacity in urban areas (Anandakumar, 1999; Grimmond and Oke, 1999; Oke et al. 1999). Dry or limited vegetation will
increase the UHII, but urban structures and materials can both shade and enhance storage heat fluxes leading to delayed
turbulent heat fluxes, decreasing the urban near-surface temperatures. Negative afternoon UHIIs previously observed in
Beijing (Wang et al. 2017) were ascribed to a substantial discrepancy between urban and rural measurement heights, but this
is not a factor here (Sect. 2.2 and 2.5).

Modelled daytime UHIIs with elevated AHEs exceed the observed values (~ 0.3 °C in winter and ~ 0.9 °C in summer,
Table 4; Fig. 6). However, previously observed daytime UHIIs in central Beijing at multiple meteorological stations (Yang et
al. 2013; Jiang et al. 2019) are of similar magnitude to the AHE50 case values. The model's inability to replicate the negative
daytime UHIIs observed at the urban site may be due to an underestimation of afternoon storage heat flux or possibly that
there are nearby fine-scale green spaces, unresolved in the land cover data implemented for this study, that increase evaporative
cooling at IAP and limit its representativeness of the central Beijing region.

Model predictions of summer afternoon UHIIs are improved when urban surface resistance to evaporation is reduced
from 200 s m$^{-1}$ to 150 s m$^{-1}$ (Evp150 case), with cooler urban air temperatures simulated as the latent heat flux (Eq. 3) is
increased ($R$ = 0.52; Fb = -0.17; Table 4). High moisture availability in central Beijing (i.e. low Bowen ratio = sensible heat
flux/ latent heat flux), and specifically at IAP, has previously been observed (Dou et al. 2019) and is thought to be related to
extensive use of water for road cleaning and for irrigating the high density of greenbelts in urban Beijing. Smaller decreases
in simulated daytime temperature in winter, after increasing urban moisture levels, are due to weak levels of incoming SW
radiation. Negligible differences (< 0.2 °C) occur in modelled winter UHIIs between AHE50 and Evp150 cases in contrast
with substantial reductions to all values (0.5-1.5 °C) in summer (Fig. 7). Furthermore, the simulated diurnal temperature range
in summer without AHEs (noAHE case) is three times higher than in winter (~ 2.5 °C) (Fig. 6), driven by much stronger
incoming SW radiation, which enhances the urban-rural differences between available energy partitioning and therefore
impacts all the surface energy balance fluxes (Eq. 3).

Differences between UHIIs simulated with and without AHEs (both cases with high urban moisture levels), quantify
the diurnally averaged and hourly AHE contributions to the UHI in winter and summer (Fig. 6 and 7). These results suggest
that Beijing's urban warming in winter is dominated by AHEs, with a small contribution from surface radiative effects alone
between -0.5 °C and +0.1 °C. In summer, AHEs increase the modelled UHII by ~60-80% throughout the day (Fig. 6), peaking
at 84 % at 8 pm, associated with heat released from residential cooling systems and rush hour traffic (Fig. 5) accumulating in
a stabilising evening PBL (Shi et al. 2019; Hertwig et al. 2020). Wang et al. (2013) simulated a similar daily maximum AHE
contribution to the UHI in Beijing of 75 %. Excluding AHEs causes winter and summer UHII biases to grow (RMSE = 4.05



and 3.07, respectively), whereas the temporal variations become more similar ($R$ = 0.62 and 0.76 in winter and summer, respectively; Table 3), reflected by the increased linearity between observed and noAHE modelled UHIIs in Fig. 7. Substantially larger error bars associated with both simulations that include AHEs are related to increased measured-modelled differences caused by simulated and real-world PBL dynamics differences and subsequent impacts on modelled heat dispersion.

Quantifying the relative importance of urbanisation-induced surface energy balance changes, including AHEs, is useful for urban planners. Most notably, our results suggest that strategies aimed at reducing the daytime storage heat flux would help to decrease nighttime UHIIs in summer by reducing nocturnal heat release, hence lowering the cooling energy demand at night and therefore the contribution from AHEs to urban warming.

### 3.2 High-resolution spatial temperature variations in summer

Maps of near-surface air temperature provide information on the location and physical characteristics of the warmest urban microclimates that may pose the biggest health risk to residents during extreme heat events. Urban cool islands (higher rural versus urban temperatures) associated with waterways and green spaces (K. Wang et al. 2019), can also be identified.

Neighbourhood-scale resolution (~ 100 m) maps of simulated air temperature (2.5 m above ground level) across urban Beijing on 23 May 2017 at 11:00 am are compared with Landsat 8 LSTs (Sect. 2.5) in Fig. 8. Near-surface temperatures are modelled with Base and Evp150 case surface resistance to evaporation values (AHEs excluded) to test the representativeness of the enhanced urban moisture scenario for the full model domain.

Generally, the range of LSTs far exceeds that of near-surface air temperatures across urban areas. Large differences in thermal properties between impervious surfaces and the atmosphere, and different heating mechanisms, cause LSTs to heat more rapidly in response to the absorption at the surface of strong incident SW radiation in summer, whereas the atmosphere heats by convection (Anandakumar, 1999). Differences between LSTs and air temperatures vary diurnally, with the Landsat overpass time (11:00 am) coinciding with an atmospheric urban cool island in summer (Fig. 6b), but a growing surface urban heat island that peaks in the early afternoon (Meng et al. 2018; Li et al. 2020). Furthermore, air temperatures within the urban canopy are impacted by all of the energy exchange processes, including micro-scale advection of heat from nearby surfaces with varying moisture content, thermal properties and aerodynamic roughness, and therefore can become uncoupled from the LSTs (Voogt and Oke, 2003). These inherent differences between the two variables limit direct comparisons, however, the relative spatial patterns are of interest.

The LSTs range between 16 and 47 °C within the domain (Fig. 8). They generally peak within the 5[th] Ring Road and decrease with distance from the urban centre as the amount of green space increases (Estoque et al. 2017). Small areas of high LSTs in suburban zones between the 5[th] and 6[th] Ring Roads to the NW and SE (Fig. 8d) are reflective of the suburban expansion during the last decade observed by Liu et al. (2020). The modelled near-surface air temperatures at 11:00 am range between 15 and 27 °C within the Sixth Ring Road (Fig. 8a and 8b). This spatial variation is greater during the early evening hours (not shown), when the canopy layer UHI peaks (Fig. 6b), with near-surface temperature hotspots more clearly defined.

Correlation between the spatial variation of LSTs and simulated near-surface air temperature improves when using the Base case surface resistance to evaporation values ($R$ = 0.58), relative to the high urban moisture case ($R$ = 0.52; Fig. 9). This is caused by modelled air temperature increases within the Fifth Ring Road of 2 to 3 °C (Fig. 8c) as a result of reduced daytime evaporative cooling. Near-surface air temperatures simulated with high urban moisture do not have the same general decrease with distance from the urban centre as seen in the LSTs (Fig. 8) because the spatial variability in the surface resistance to evaporation has been removed. This is reflected in Fig. 9b, with no relationship between the highest LSTs and Evp150 case simulated air temperatures. Equal urban and upwind surface resistance to evaporation values (150 s m⁻¹) lead to simulated central urban air temperatures that are mainly controlled by the much greater thermal admittance values across the modelling domain versus the upwind domain. This lowers central urban near-surface air temperatures at 11:00 am, relative to suburban



regions between the Fifth and Sixth Ring Roads that have lower thermal admittances (Fig. 2), producing a homogeneous urban near-surface air temperature distribution (Fig. 8b). This is further indication that local surface radiative cooling, likely due to nearby green spaces, differentiate the urban site from the mean conditions across the central Beijing region. Worse spatial correlation with LSTs, but better agreement with near-surface air temperature measurements, with increased urban moisture also reflects expected differences between LSTs and near-surface air temperatures. Micro-scale advection and turbulent diffusion within the urban canopy can mix warmer and cooler pockets of air, reducing the coupling between LSTs and air temperatures (Roth et al. 1989).

The lowest LSTs and modelled air temperatures correlate most strongly (Fig. 9), corresponding with green spaces, waterways (Fig. 2) and areas of high elevation (Fig. 3). The model's ability to capture these fine-scale urban cool islands, such as Qianhai Lake near the centre of Beijing (Fig. 2), in addition to the general urban temperature pattern, highlights the successful implementation for this study of the LCZ and OSM data. Increased correlation between modelled near-surface air temperatures and LSTs may require more detailed sub-divisions of urban land cover, such as specific information on building façade or roof materials as opposed to general descriptions of building function (Aktas et al. 2017).

### 3.3 Spatiotemporal UHI variations near the airport

Large-scale urban developments strongly impact local air temperatures, following the replacement of vegetative surfaces with expanses of concrete and asphalt, and therefore affect the thermal comfort, cooling and heating energy demand, and air quality across neighbouring residential areas (Hamilton et al. 2014). In this section, we investigate the extent to which simulated spatial temperature patterns near Beijing Capital International Airport vary throughout the day due to different surface characteristics. Figure 10 shows a satellite image of the airport and its surroundings alongside maps (100 m resolution) of the surface properties that have the greatest influence on modelled near-surface air temperatures (CERC, 2018).

Figure 11 presents modelled local temperature perturbations (excluding AHEs) relative to upwind values, across the region displayed in Fig. 10, at four different times of the day on 23 May 2017. At 05:00, prior to sunrise and in the absence of incoming SW radiation, modelled near-surface air temperatures at the airport are minimally affected by the underlying impervious surface (Fig. 11a). Cool regions adjacent to the Wenyu River, running SE to NW (Fig. 10), with UHIIs of ~ -2 to -3 °C are related to green spaces with both low thermal admittance and surface resistance to evaporation.

At 14:00, when the mean canopy layer UHII at the urban site peaks (Fig. 6b), a clear spatial temperature pattern has developed (Fig. 11b). Modelled UHIIs at the airport site rise to ~6-8 °C caused by the high resistance to evaporation (200 s m$^{-1}$) and low albedo (0.08) of the airport runways (Table A1). The low albedo surface absorbs more of the strong daytime incoming SW radiation, with a greater proportion of the net radiation partitioned to the upwards convective flux of sensible heat due to the low surface moisture content. The fine-scale structure of the Wenyu River is clearly defined at 14:00 as the water's near-zero resistance to evaporation cools near-surface temperatures relative to the neighbouring vegetation.

After sunset, at 19:00, UHIIs to the north and west of the airport peak at ~7-9 °C, matching air temperatures at the airport (Fig. 11c). The higher thermal admittance of these areas, relative to the airport, causes the release of greater amounts of stored heat in the early evening. Dispersion of this upwelling ground heat flux is inhibited by a stabilising nocturnal PBL, creating a positive temperature increment to the north of ~ 2 °C, relative to the cooling airport region, at 23:00 (Fig. 11d).

Figure 12a shows the land cover at the residential (2) and forest (3) sites, marked in Fig. 10 and Fig. 11. The sites are only ~ 1 km apart, either side of the Wenyu River, but experience their own distinct microclimates driven by the underlying land surface characteristics. This is quantified in Fig. 12b by the mean summer diurnal modelled near-surface air temperature differences between the airport site and both the residential and forest sites.

After sunrise, the temperature difference (ΔT) between the airport and both the residential and forest sites increases (Fig. 12b). ΔT rises more sharply at the forest site, due to its lower surface resistance to evaporation and therefore stronger evaporative cooling than in the residential area, and peaks at 14:00 when the airport is ~ 3 °C and ~ 5 °C warmer than the



residential and forest sites, respectively (Fig. 12b). In the evening, after sunset, ΔT decreases to ~ 0.25 °C and ~ 0.5 °C at the residential and forest sites (Fig. 12b), respectively, as spatial temperature variations due to surface characteristics reach a minimum (Fig. 11d). These negligible modelled nocturnal temperature differences (excluding AHEs) again highlight the dominant contribution from AHEs to UHIIs at night (Fig. 6).

### 3.4 Impact of heatwaves on Beijing's UHI

Synergies between extreme heat events and UHIIs in Beijing are well-documented (Li et al. 2015; Chen et al. 2018; Jiang et al. 2019; He et al. 2020). Two heatwave periods during the summer are identified from (a) 18 May to 21 May, and (b) 14 June to 20 June (Fig. 13), with heatwave events defined here as three or more consecutive days with a daily maximum temperature ($T_{max}$) at the urban site reaching at least 35 °C, following the Chinese Meteorology Administration heatwave definition (Tan et al. 2007; Jiang et al. 2019). In Fig. 13, red shaded regions mark hours with higher urban than rural temperatures and highlight the build-up of heat in urban Beijing during the evening. Blue shaded regions mark hours with higher rural than urban temperatures and emphasise the negative summer daytime UHI (Fig. 6b).

Mean measured and modelled (Evp150 case) daytime (10:00 to 16:00) and nighttime (22:00 to 04:00) UHIIs for heatwave and non-heatwave days are presented in Fig. 14. Day and night hours are defined based on previous UHI-heatwave studies (Wang et al. 2017; Jiang et al. 2019).

At night, the mean measured UHII is 5.3 °C on heatwave days, 1.1 °C higher than on non-heatwave days. Previous nocturnal UHII increments of similar magnitude during heatwaves in Beijing (Jiang et al. 2019) have been attributed to greater daytime storage heat flux, from stronger SW and LW radiation, and a therefore more prolonged period of evening heat release warming the near-surface (Li et al. 2015). Warmer nights lead to increased residential AHEs from air conditioning units, further strengthening the UHII (Zhao et al. 2018).

Modelled UHIIs are underestimated by 31 % compared to measurements during non-heatwave nights. However, this increases to 56 % during heatwave nights and further suggests that the model-calculated evening stored heat release is too low (Sect. 3.1). Elevated daytime storage heat flux during heatwaves increases the model's underestimation of ground heat release at night, hence the measured-modelled evening UHII discrepancy grows (Fig. 14). Furthermore, the model does not account for the day-to-day accumulation of stored heat during a heatwave period, only the diurnal variation, therefore the model's underestimation of evening heat release grows incrementally throughout a heatwave. The use of the same diurnal AHE profile for all days also likely contributes, with the proportion of AHEs released at night expected to increase as cooling energy demand at night grows during heatwaves (Sailor and Vasireddy, 2006).

During the day, the mean measured UHII is -1.34 °C on heatwave days, compared to -0.57 °C on non-heatwave days (Fig. 14). This is contrary to previous measurements of Beijing's daytime UHII during extreme heat events which have shown an increase relative to normal days (Li et al. 2015; He et al. 2020) explained by (a) greater urban-rural evapotranspiration and sensible heat flux differences, (b) stable synoptic conditions associated with heatwaves reducing mixing between urban and rural air, and (c) elevated AHEs (Li and Bou-Zeid, 2013; Li et al. 2015; He et al. 2020). Therefore, the daytime heatwave UHII decrease observed at the urban site is likely another example of the impact of local site characteristics (e.g. nearby vegetation) on air temperatures at IAP. Overall, extreme temperature events clearly worsen measured-modelled UHII comparisons in summer and model performance is substantially better under non-heatwave conditions.

### 4    Conclusions

We use the urban climate component of the ADMS-Urban model to investigate the spatiotemporal variations of Beijing's canopy layer UHI. The diurnally varying contributions from land surface characteristics and anthropogenic heat emissions (AHEs) to urban heat island intensities (UHIIs), during winter and summer periods, are examined through four different



simulations. Neighbourhood-scale resolution (100 m) maps of simulated near-surface air temperature within Beijing's Sixth Ring Road are evaluated against satellite-derived LSTs. Interactions between heatwaves and Beijing's UHI are also explored. Beijing's winter UHI is dominated by AHEs, with negligible UHIIs simulated due to land surface radiative effects alone.

Modelled nocturnal UHIIs are underestimated by ~ 2 °C compared with measurements at the urban meteorological site, located at the Institute of Atmospheric Physics (IAP), despite allowing for an increase of 50 % in AHEs. This suggests that the urban site is impacted by strong, local AHE sources (e.g. adjacent busy roads or residential buildings) that are unresolved by the aggregate 3 km AHEs generated for this study. Development of higher-spatially resolved AHEs (< 1 km) is recommended to fully capture Beijing's winter UHI in future work.

Modelled nocturnal UHII underestimations in summer relative to measured values at the urban site, which increase by ~ 25 % during heatwave days, are likely related to evening ground heat flux values simulated by ADMS-Urban that are too low due to restrictions on modelled upwind PBL stability. This effect is more pronounced during heatwaves as the accumulation of stored heat throughout extreme temperature events is not accounted for by the model and the diurnal AHEs profile used for this study is not adjusted to capture the likely greater proportion of AHEs released at night during heatwaves by residential cooling systems. Future modification of the model's calculation of ground heat release is recommended to better

represent the prolonged evening turbulent transfer of stored heat and replicate the large nocturnal UHIIs across parts of Beijing and other densely built megacities.

     Increased urban moisture levels improve the agreement between simulated and measured daytime UHII values in summer at the urban site. By reducing surface resistance to evaporation values from 200 s m$^{-1}$ to 150 s m$^{-1}$ across central urban areas, thereby enhancing daytime evaporative cooling, daytime measured-modelled UHII agreement greatly improves (Fb

decrease from 1.26 to -0.17 and $R$ value increase from 0.31 to 0.52). However, the spatial correlation between LSTs and modelled near-surface air temperatures across the domain is stronger with the Base case model configuration. This suggests that conditions at the urban site are impacted by evaporative cooling from local, fine-scale features that are not captured by the land use data and highlights the expected impact of small-scale advection within the urban canopy that can uncouple near-surface air temperatures and LSTs. Further improvement to simulated spatial temperature variations at high-resolution requires

more refined urban land use information (e.g. building materials) to enhance modelled temperature heterogeneity across central Beijing.

     The implementation of LCZ and OSM land use information for this neighbourhood-scale urban climate study has, however, generally proven successful, particularly in capturing cooler regions associated with green spaces, waterways and areas of high elevation. The benefits of urban planning strategies aimed at increasing the density of these cooler spaces are

highlighted by an analysis of the local climate impact of Beijing Capital International Airport, revealing that typical daytime temperatures in summer can be up to ~ 5 °C warmer than nearby green spaces.

     This study provides critical information for urban planners on the key processes impacting Beijing's canopy layer UHI. Reducing the heat-related mortality risk through effective UHI mitigation strategies in megacities such as Beijing is of ever-increasing importance and urgency in our warming climate. The methodologies and open data sources applied here

provide a framework that future neighbourhood-scale urban climate modelling studies, in other megacities with similarly limited data availability, can build on.



**Appendix A**

**Table A1: OpenStreetMap land use types (OSM, 2020) and surface parameters (thermal admittance, surface resistance to evaporation and albedo) based on literature-reported values (Oke, 1982; Cox et al. 1999; Hamilton et al. 2014; CERC, 2018). Land use types are allocated to urban, green or water categories shown in Fig. 2 and Fig. 12.**

| OSM land use type | Thermal admittance ($J K^{-1} m^{-2} s^{-1/2}$) | Surface resistance to evaporation (s/m) | Albedo |
|---|---|---|---|
| Urban | | | |
| Residential | 1500 | 150 | 0.18 |
| Industrial | 1750 | 200 | 0.12 |
| Commercial | 1500 | 200 | 0.12 |
| Railway | 1150 | 200 | 0.08 |
| University | 1500 | 200 | 0.12 |
| School | 1500 | 200 | 0.12 |
| Substation | 1500 | 200 | 0.08 |
| Parking | 1205 | 200 | 0.08 |
| Fuel | 1205 | 200 | 0.08 |
| College | 1500 | 200 | 0.12 |
| Aerodrome | 1205 | 200 | 0.08 |
| Train station | 1500 | 200 | 0.12 |
| Hospital | 1500 | 200 | 0.12 |
| Terminal | 1500 | 200 | 0.12 |
| Plant | 1500 | 200 | 0.12 |
| Pedestrian | 1096 | 200 | 0.08 |
| Sports centre | 1500 | 200 | 0.12 |
| Place of worship | 1500 | 200 | 0.12 |
| Barracks | 1500 | 200 | 0.08 |
| Platform | 1205 | 200 | 0.08 |
| Stadium | 1500 | 200 | 0.12 |
| Apartments | 1500 | 200 | 0.12 |
| Cinema | 1500 | 200 | 0.12 |
| Retail | 1500 | 200 | 0.12 |
| Generator | 1500 | 200 | 0.12 |
| Quarry | 2220 | 200 | 0.08 |
| Green | | | |
| Park | 600 | 115 | 0.2 |
| Farmland | 1500 | 50 | 0.17 |
| Forest | 1400 | 115 | 0.16 |
| Wood | 1400 | 115 | 0.16 |
| Cemetery | 600 | 70 | 0.2 |
| Common | 600 | 70 | 0.2 |
| Grass | 600 | 70 | 0.2 |
| Village green | 600 | 70 | 0.2 |
| Garden | 600 | 60 | 0.19 |
| Pitch | 600 | 70 | 0.19 |
| Meadow | 600 | 70 | 0.19 |
| Scrub | 600 | 70 | 0.17 |
| Water | | | |
| Water | 1545 | 10 | 0.08 |
| Riverbank | 1545 | 10 | 0.08 |
| Reservoir | 1545 | 10 | 0.08 |

*Author contributions.* MB set up and ran the model with support from JS. MB processed the model outputs with support from JS. This paper was written by MB with guidance from JS, RMD, OW, DC and SG. All the authors read and improved the manuscript. YH, PF, SG and SK provided measurement data.

*Competing interests.* Jenny Stocker works for and David Carruthers is a director of CERC, who develop and license the ADMS-Urban model.



*Acknowledgements*. This work was funded by the UK Natural Environment Research Council (NERC) Industrial studentship scheme with CASE support provided by Cambridge Environmental Research Consultants (CERC). We would also like to acknowledge the APHH-China programme.

*Financial support*. This research has been supported by the Natural Environment Research Council (grant nos. NE/N007794/1, NE/N006941/1, NE/N006925/1, and NE/N006976/1).

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




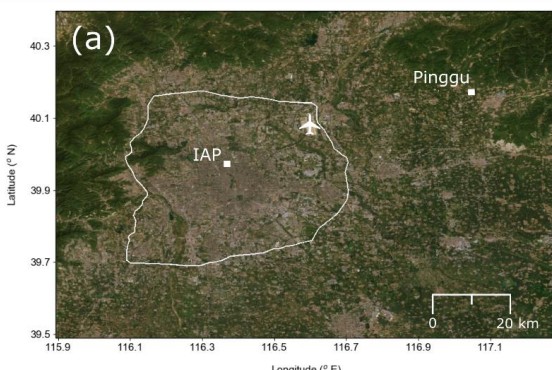

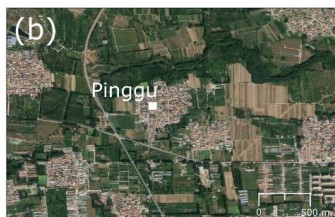

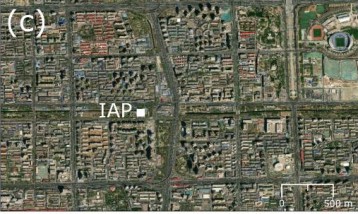

**Figure 1. Beijing area and surroundings showing locations of (a) Beijing Capital International Airport Meteorology Observatory (airplane), (b) urban meteorological site at the Institute of Atmospheric Physics (IAP), (c) rural meteorological site in Pinggu and Sixth Ring Road (white line) marking the model domain. Source of satellite imagery: Esri World Imagery.**



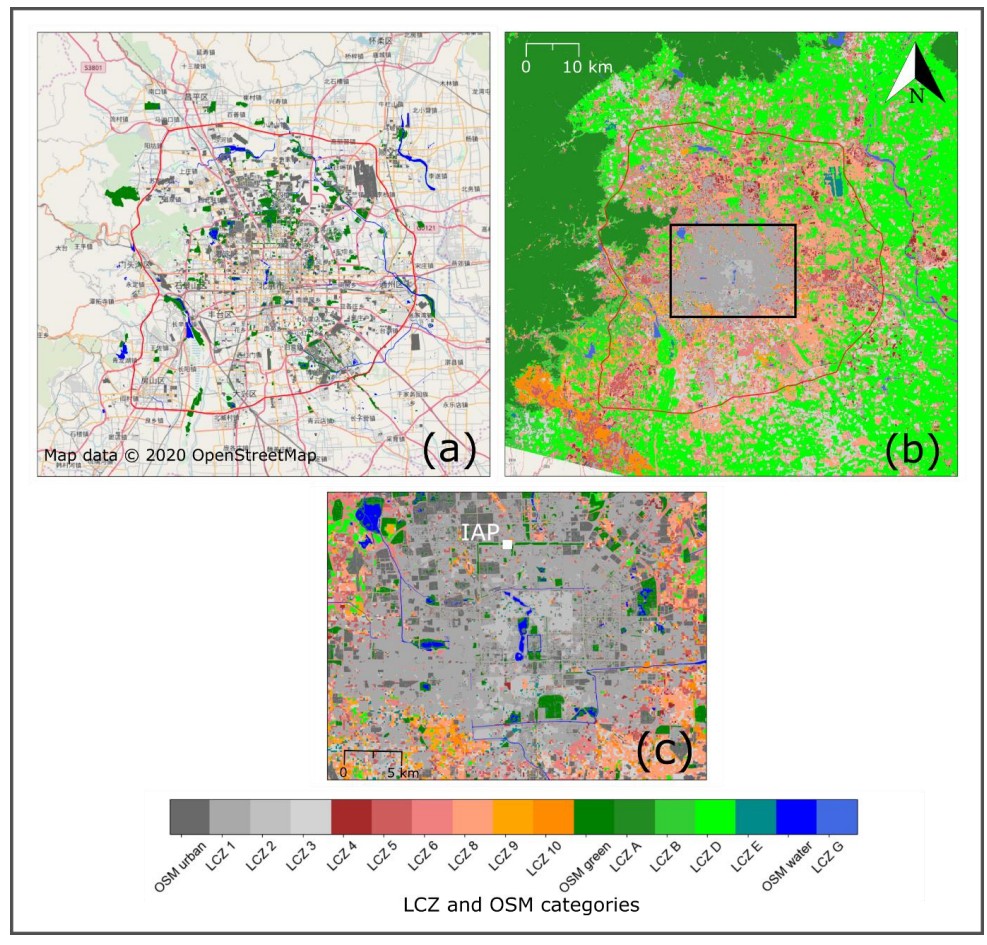

**Figure 2. Sources of data used to determine model surface parameters for Beijing, covering the model domain within the Sixth Ring Road (red line), including (a) OpenStreetMap (OSM, 2020) land use (Table A1) and (b) local climate zones (LCZ) (WUDAPT, 2020). OSM land use overlaid onto LCZs across central Beijing (black square in b) is shown in (c), with the urban meteorological site at the Institute of Atmospheric Physics (IAP) also marked. © OpenStreetMap contributors 2020. Distributed under a Creative Commons BY-SA License.**



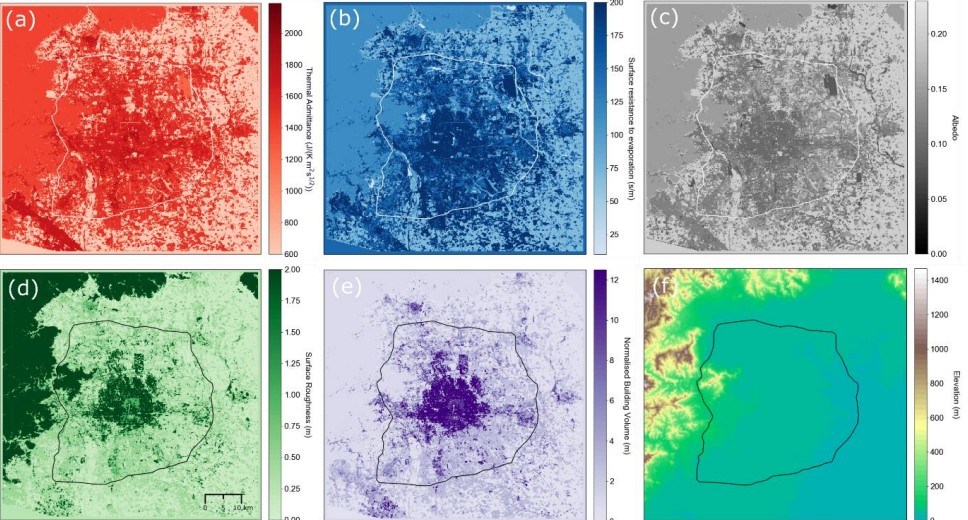

**Figure 3. Surface characteristics in Beijing area (100 m resolution) with Sixth Ring Road (white or black line), including (a) thermal admittance (J K$^{-1}$ m$^{-2}$ s$^{-1/2}$), (b) surface resistance to evaporation (s m$^{-1}$), (c) albedo, (d) surface roughness length (z$_0$) (m), (e) normalised building volume (NBV) (m), and (f) terrain elevation (USGS, 2020). See Sect. 2.3 for data sources and method describing how parameter values were determined.**


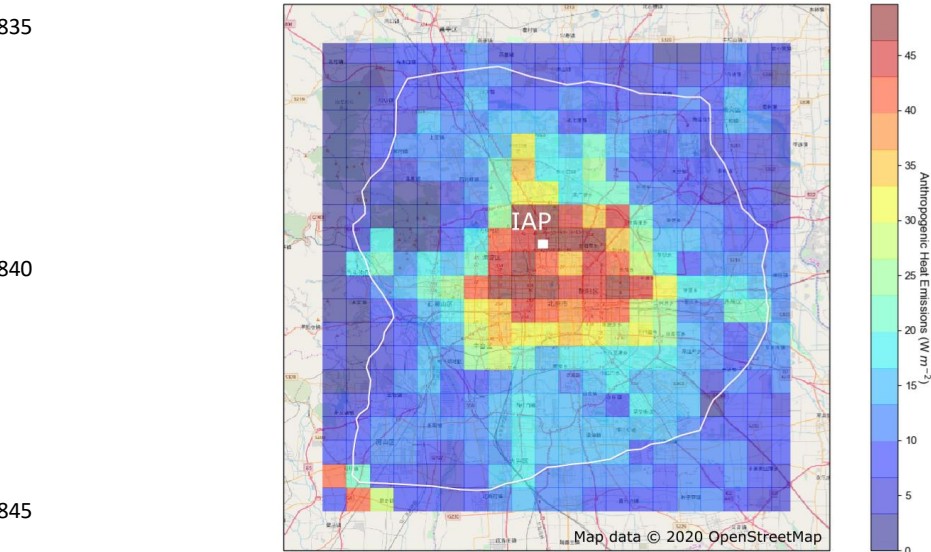



**Figure 4. Aggregate anthropogenic heat emissions (AHE) (3 km resolution) across Beijing. Magnitudes based on Dou et al. (2019) with spatial weightings based on the local climate zone (LCZ) locations (Fig. 2) and Stewart and Oke's (2012) AHE values for each LCZ classification. Sixth Ring Road and urban meteorological site (IAP) are marked. © OpenStreetMap contributors 2020.**
**Distributed under a Creative Commons BY-SA License.**








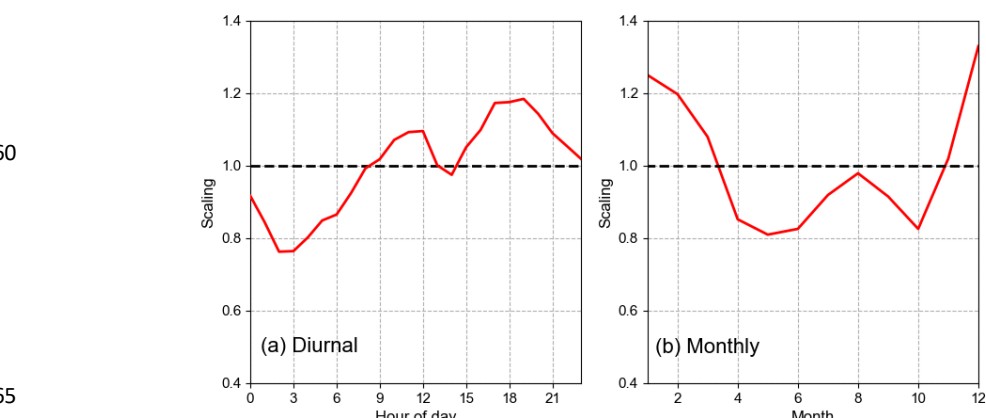

Figure 5. (a) Diurnal and (b) monthly anthropogenic heat emission (AHE) profiles. See Sect. 2.4 for method describing how profiles were determined.




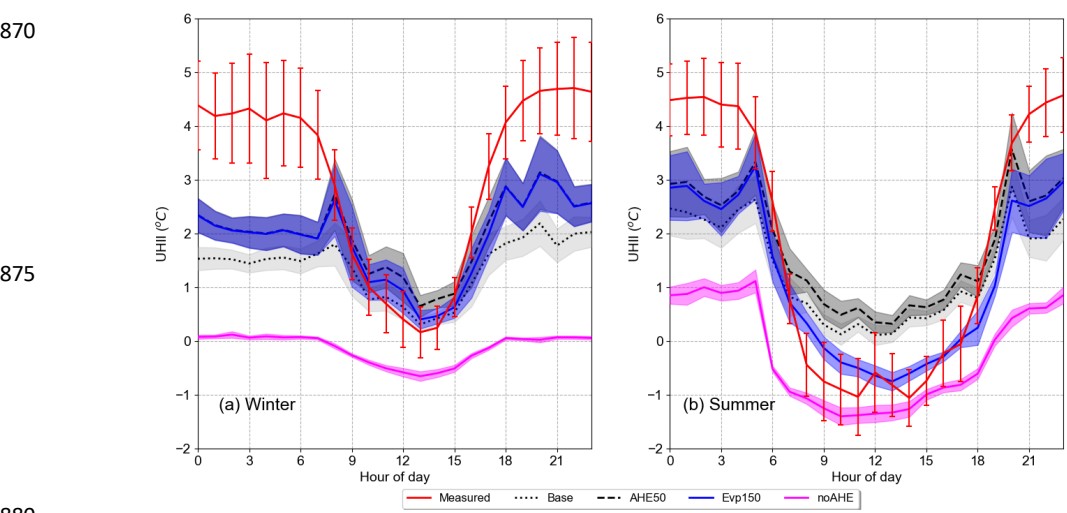

Figure 6. Mean diurnal variation in measured and modelled urban heat island intensities (UHIIs) at the urban site in (a) winter and (b) summer. Modelled UHIIs from Base (black dotted), AHE50 (black dashed), Evp150 (blue) and noAHE (pink) cases are presented. Measurements are marked by the red line. Shaded regions and error bars represent the 95 % confidence intervals for modelled and measured UHIIs, respectively.







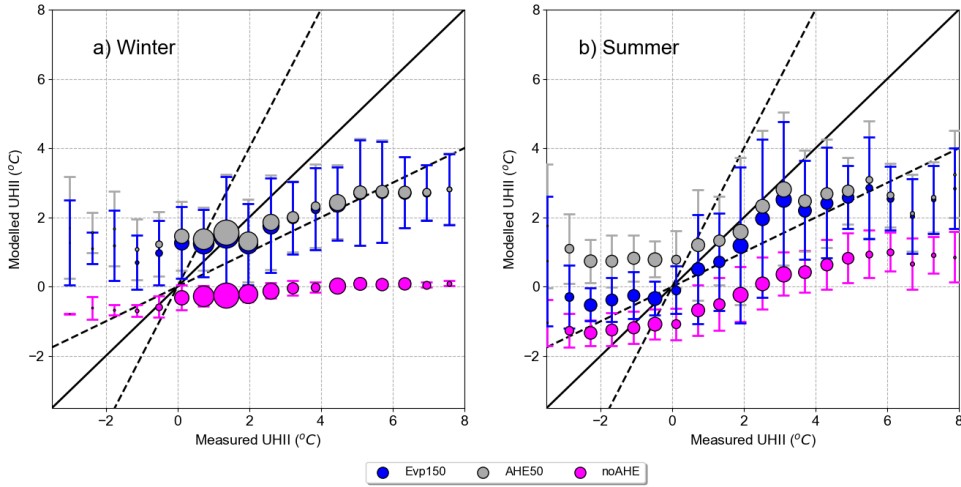

**Figure 7. Hourly measured and modelled urban heat island intensities (UHIIs) for (a) winter and (b) summer periods. Modelled UHIIs from Evp150 (blue), AHE50 (grey) and noAHE (pink) cases are presented. Measured UHIIs are grouped into bins (0.5 °C), points representing the mean modelled UHII in the bin. Point sizes scaled by total number of hourly values per bin. Error bars represent 1 standard deviation of hourly modelled UHIIs in each bin.**





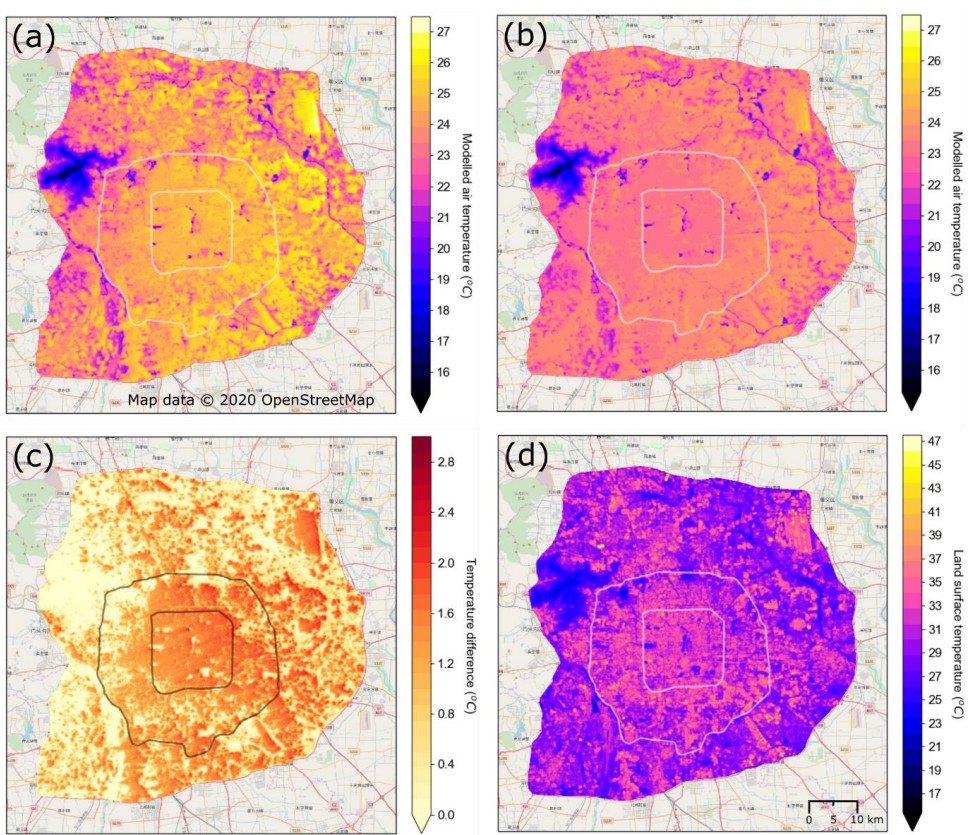

**Figure 8. Near-surface (2.5 m above ground level) air temperatures and land surface temperatures (LST) within Beijing's Sixth Ring Road on 23 May 2017. Air temperatures modelled at 11:00 am, excluding anthropogenic heat emissions, with surface parameters from (a) Base, and (b) Evp150 cases. (c) Differences between air temperatures modelled in (a) and (b) (a minus b). (d) Landsat 8-derived LSTs (USGS, 2020) at 10:53 am. Beijing's Fifth (outer) and Third (inner) Ring Roads marked by white (a, b and d) and black (c) lines. © OpenStreetMap contributors 2020. Distributed under a Creative Commons BY-SA License.**






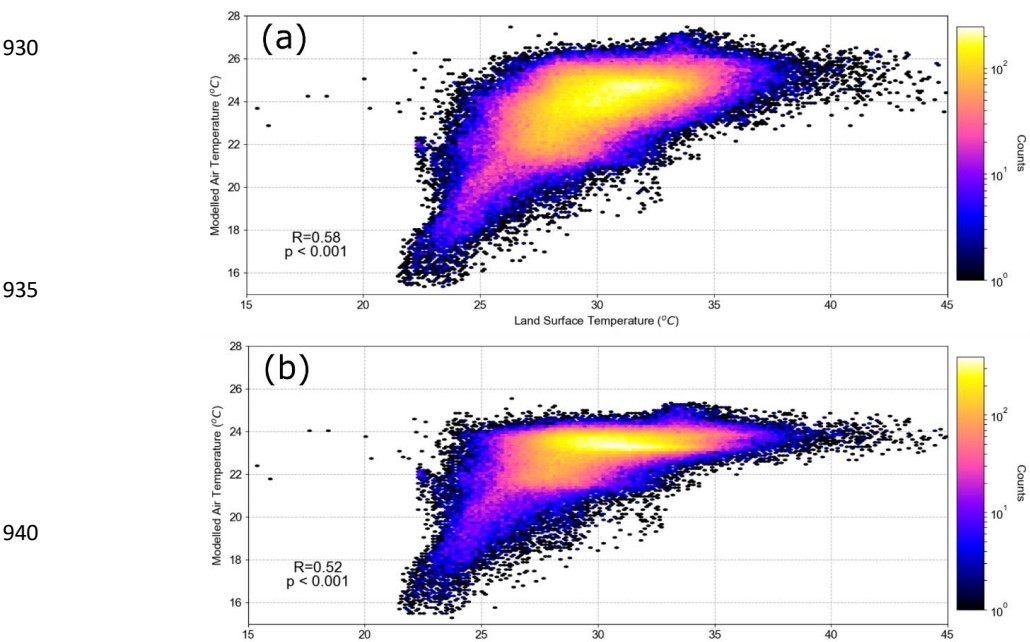


**Figure 9. Frequency (colour) of 100 m resolution Landsat 8-derived land surface temperatures (USGS, 2020) and near-surface (2.5 m above ground level) air temperatures, modelled with surface parameters from (a) Base, and (b) Evp150 cases, both excluding anthropogenic heat emissions.**



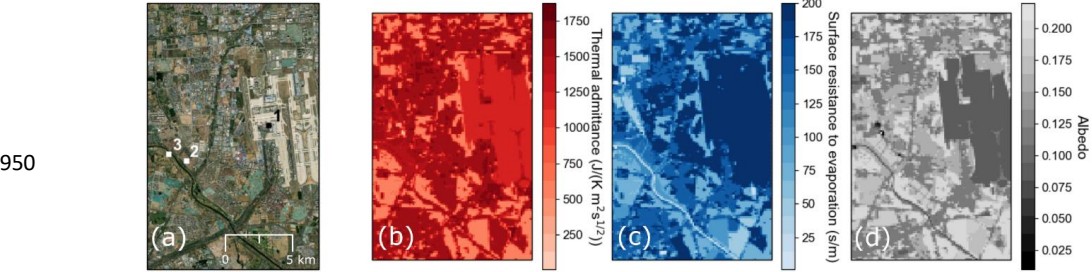

**Figure 10. Beijing Capital International Airport and its surroundings (a) (source of satellite imagery: Esri World Imagery), with locations of the airport (1), a residential area (2) and a forested region (3) indicated. Base case (b) thermal admittance, (c) surface resistance to evaporation and (d) albedo values (100 m resolution) within the region shown in (a).**






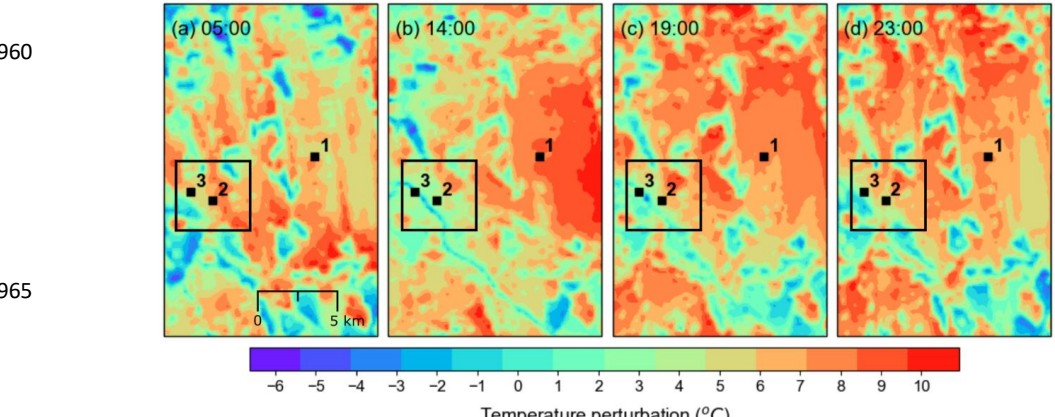

**Figure 11. Modelled urban heat island intensities using Base case surface parameters, excluding anthropogenic heat emissions, within the region shown in Fig. 10 at (a) 05:00, (b) 14:00, (c) 19:00, and (d) 23:00 on 23 May 2017. Airport (1), residential (2) and forest (3) sites are marked. Square marks area covered in Fig. 12a.**

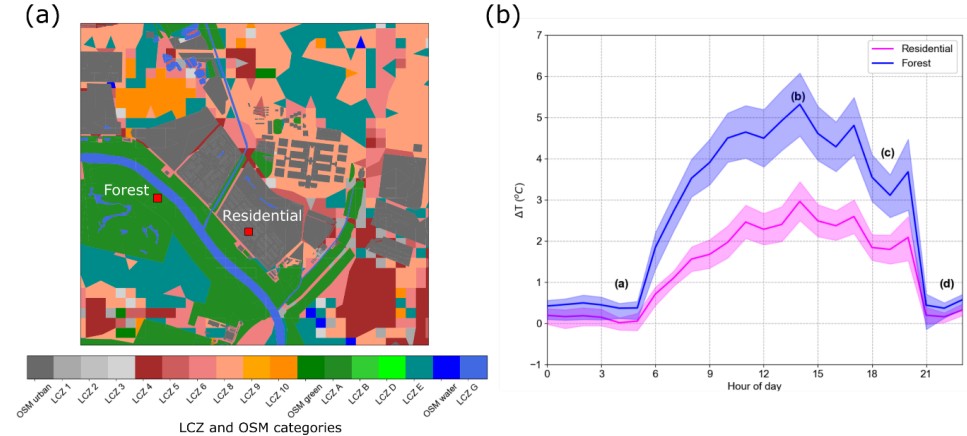

**Figure 12. (a) Local climate zones (WUDAPT, 2020) combined with OpenStreetMap (OSM, 2020) data within the region marked in Fig. 11. See Fig. 11 for locations of Residential (2) and Forest (3) sites. (b) Mean summer diurnal modelled near-surface (2.5 m above ground level) air temperature differences (ΔT) between the airport and (pink) residential, and (blue) forest locations. (a), (b), (c) and (d) marked in panel (b) correspond to the times of each modelled near-surface (2.5 m) air temperature map in Fig. 11. Shaded regions represent the 95 % confidence intervals for ΔT values. © OpenStreetMap contributors 2020. Distributed under a Creative Commons BY-SA License.**





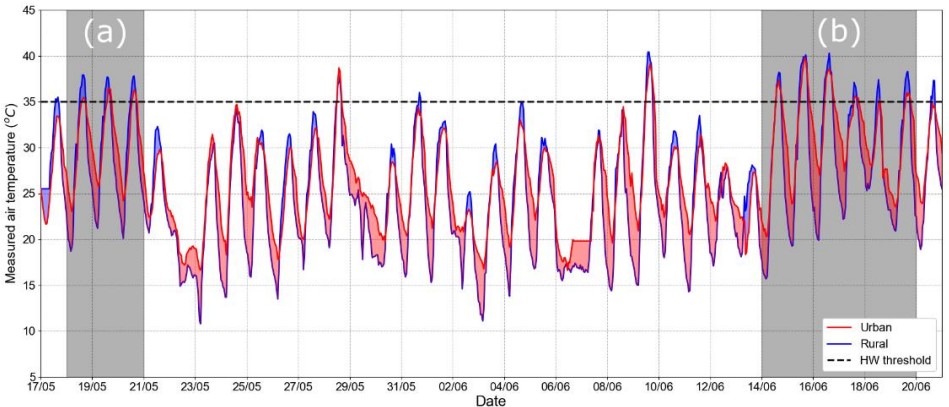

**Figure 13. Hourly measured air temperatures (~ 8 m above ground level) at the urban (red) and rural (blue) sites during the summer period (17 May – 21 June 2017). Two heatwave events (shaded grey regions) are highlighted from (a) 18 May to 21 May, and (b) 14 June to 20 June 2017. Daily maximum temperature threshold for heatwave definition is also marked (black dashed line).**

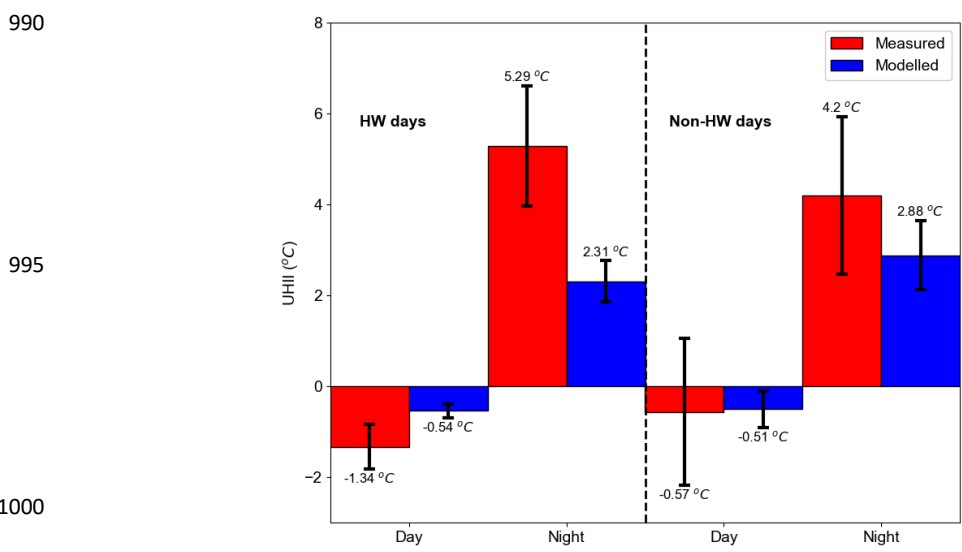

**Figure 14. Mean measured (red) and Evp150 case modelled (blue) urban heat island intensities (UHIIs) for heatwave (HW) and non-heatwave (non-HW) days (Fig. 13) for day (10:00 to 16:00) and night (22:00 to 04:00) hours. Error bars represent 1 standard deviation of daily mean day and night UHIIs. Mean UHIIs for each period are marked either above or below error bars.**





**Table 1. Surface parameter values assigned to Local climate zones (LCZ) mapped in Beijing (WUDAPT, 2020). Values, except for surface resistance to evaporation, from Stewart and Oke (2012). See Sect. 2.3 for method describing how values are chosen from their ranges. Surface resistance to evaporation is based on literature-reported values (Oke, 1982; Cox et al. 1999; Hamilton et al. 2014; CERC, 2018).**

| LCZ | Description | Thermal admittance $(J\ K^{-1}m^{-2}s^{-1/2})$ | Albedo | Surface resistance to evaporation (s m$^{-1}$) | $z_0$ (m) | NBV (m) |
|---|---|---|---|---|---|---|
| 1 | Compact high-rise | 1650 | 0.1 | 200 | 2 | 12.5 |
| 2 | Compact midrise | 1850 | 0.1 | 200 | 1 | 9.625 |
| 3 | Compact low-rise | 1500 | 0.1 | 200 | 0.5 | 3.575 |
| 4 | Open high-rise | 1600 | 0.12 | 150 | 2 | 7.5 |
| 5 | Open midrise | 1700 | 0.12 | 150 | 0.5 | 5.25 |
| 6 | Open low-rise | 1500 | 0.12 | 150 | 0.5 | 1.95 |
| 8 | Large low-rise | 1500 | 0.15 | 200 | 0.25 | 2.6 |
| 9 | Sparsely built | 1400 | 0.12 | 150 | 0.25 | 0.975 |
| 10 | Heavy industry | 1750 | 0.12 | 200 | 0.25 | 2.5 |
| A | Dense trees | 1400 | 0.15 | 115 | 2 | 0.825 |
| B | Scattered trees | 1400 | 0.2 | 115 | 0.5 | 0.45 |
| D | Low plants | 600 | 0.2 | 70 | 0.1 | 0.025 |
| E | Bare rock or paved | 1850 | 0.225 | 200 | 0.005 | 0.00625 |
| G | Water | 1545 | 0.06 | 10 | 0.0002 | 0 |
| Upwind domain | | | | | | |
| Rural site (LCZ 9 modified) | 600 | 0.2 | 150 | 0.25 | 0.975 |


**Table 2. Surface parameters and anthropogenic heat emissions (AHEs) used in the four model experiments.**

| Case | Surface parameters | AHEs |
|---|---|---|
| Base | As Table 1 and Table A1 | As Fig. 4 and Fig. 5 |
| AHE50 | Base | Base + 50 % |
| Evp150 | Base with Evp (LCZ 1-3, 8, 10 and urban OSM) = 150 s m$^{-1}$ | Base + 50 % |
| noAHE | Base with Evp (LCZ 1-3, 8, 10 and urban OSM) = 150 s m$^{-1}$ | 0 |


**Table 3. Statistical evaluation of modelled urban heat island intensities (UHIIs) at the urban site, for all hours of the day, during the winter (W) and summer (S) periods. Model experiments are described in Sect. 2.5. Mean UHIIs and statistics are calculated from matching hourly values.**


| UHII (°C) | | | | | Model evaluation statistics | | | | | |
|---|---|---|---|---|---|---|---|---|---|---|
| Observed | | Modelled | | | RMSE (°C) | | Fb | | *R* | |
| W | S | Case | W | S | W | S | W | S | W | S |
| 3.1 | 1.8 | Base | 1.4 | 1.3 | 2.90 | 2.66 | -0.76 | -0.30 | 0.47 | 0.48 |
| | | AHE50 | 2.0 | 1.8 | 2.69 | 2.69 | -0.43 | -0.02 | 0.39 | 0.44 |
| | | Evp150 | 1.9 | 1.2 | 2.68 | 2.49 | -0.48 | -0.44 | 0.43 | 0.59 |
| | | noAHE | -0.1 | -0.2 | 4.05 | 3.07 | -2.00 | -2.00 | 0.62 | 0.76 |


**Table 4. Statistical evaluation of modelled urban heat island intensities (UHIIs) for the winter (W) and summer (S) periods, as in Table 3, for daytime (D) (simulated K$^+$ > 0) and nighttime (N) hours.**




| | UHII (°C) | | | | | | | | Model evaluation statistics | | | | | | | | | | | |
| | Observed | | | | Modelled | | | | RMSE (°C) | | | | Fb | | | | R | | | |
| | W | | S | | W | | S | | W | | S | | W | | S | | W | | S | |
| Case | D | N | D | N | D | N | D | N | D | N | D | N | D | N | D | N | D | N | D | N |
| Base | 1.1 | 4.3 | 0.2 | 4.3 | 0.9 | 1.7 | 0.7 | 2.3 | 1.62 | 3.45 | 2.27 | 3.19 | -0.24 | -0.86 | 1.01 | -0.61 | 0.31 | 0.30 | 0.31 | -0.01 |
| AHE50 | | | | | 1.4 | 2.4 | 1.1 | 2.9 | 1.80 | 3.11 | 2.41 | 3.10 | 0.21 | -0.58 | 1.26 | -0.40 | 0.30 | 0.22 | 0.31 | -0.04 |
| Evp150 | | | | | 1.2 | 2.3 | 0.2 | 2.7 | 1.71 | 3.12 | 1.99 | 3.14 | 0.04 | -0.59 | -0.17 | -0.46 | 0.36 | 0.22 | 0.52 | -0.03 |
| noAHE | | | | | -0.4 | 0.1 | -0.9 | 0.8 | 2.13 | 4.85 | 2.25 | 4.05 | -2.00 | -1.95 | -2.00 | -1.39 | 0.57 | 0.20 | 0.61 | 0.29 |