# Peer review of "Modelling spatiotemporal variations of the canopy layer urban heat island in Beijing at the neighbourhood-scale"

_Atmospheric Chemistry and Physics, 2020_

## Referee Comment (RC1) · Anonymous Referee #2 · 27 Nov 2020

General Comments

The paper entitled "Modelling spatiotemporal variations of the canopy layer urban heat island in Beijing at the neighbourhood-scale" uses the ADMS-Urban climate model to depict the influence of the Beijing metropolitan area on the climate. Although this paper could be of substantial added value for the community, in particular by highlighting the strength of the ADMS-Urban model in comparison to other models, or also by deeply studying the biases during heat wave conditions, the paper sticks to a rather descriptive tone that does not seem to answer any research question. I have serious concerns about some methodological aspects that are given below as major comments. In par-

ticular the drawing of some conclusions based on one satellite image only cannot be considered as a strong evidence to sustain some arguments given by the authors. I would recommend improving the introduction to define the key research questions that will be answered in the paper. The methodological section will also need to be re-worked to be more easily understood. In the results, there are numerous parts that could be implemented in the discussion part, depending on the research questions to be answered. Finally, I would like the authors to clearly enumerate the key outcomes of their research in the conclusions, their limitations, and potential guidelines to continue their effort.

Major Comments

- Lines 70 - 79: Could you add something about the temporal coverage of satellites? Afterall, most of them only take snapshots of the SUHII and this further limits their use to understand when people may be at higher risks during the day.

- Lines 103 - 104: There are many other works which implemented LCZ in regional climate models to study the UHI. Actually, the work from Alexander et al. (2015) used the SUEWS model that you describe as an urban energy balance model, while your model is an urban climate model. Also, I do not think that Alexander et al. (2015) used OSM for making their LCZ map. Please clarify these points here to make clearer on what your study "builds upon".

- At the end of the Introduction I am missing some key research questions which you will try to answer. Your description on the challenges related to the use of remote sensing and in-situ measurements are properly highlighted in a very clear and efficient manner as well as the use of UCMs for coping with these limitations. But I don't really get what your study will bring further. Could you add a paragraph on that before detailing the contents of the subsequent section? Also, could you explain why you chose the ADMS-Urban model in particular? This could be explained in Section 2.1 too.

- Line 127: What do you define as "upwind"? How do you get this information since it

is at the baseline of your model from what I understand. I am not really familiar with the ADMS-Urban climate model and I miss a bit how it actually integrates information at the boundaries to the inner parts of the domain.

- Lines 160 - 174: Here you say that the model is driven by rural stations. Are they at the boundary of your domain? You use only the Airport weather station to force your model from what I understand. What are the potential limitations to that for an extended city as Beijing? Did you also consider that the airport is located in a rather built up environment (when looking at your Fig.1)? Why not using the Pinggu Station instead? Also, do we have to understand that the winter and the summer domains are located at the same place based on the winter wind direction? In general I had a hard time understanding which data is used to drive the model, what is your domain extension, its horizontal resolution and why you chose those three stations among others.

Lines 191 - 195: How did you get these values of 200 and 150? What is the rationale for saying that one is representative of densely built-up LCZ and the other of more open urban typologies? Please explain.

Lines 200 - 211: This paragraph partially answers some of my previous comments but I still have a very hard time understanding what is "upwind". If it is a border surrounding a domain then it means that all winds converge to the city center? In general I think that the methodology part will need a complete restructuring.

Lines 219 - 221: Please explain how this is spatialized throughout the domain and provide the reader with an equation on the scaling from one LCZ to another.

- Section 2.5 is among my main concerns in the methodology. I somewhat understand the reasoning behind those choices but I don't think that using LST to evaluate air temperature is a good thing to do. Additionally, the use of only one Landsat 8 image for evaluating a model that is run for 2 months seems really limited to me. To that I would like to add that Landsat 8 LSTs retrieved by Jimenez and Muñoz techniques are directly correlated to the NDVI. The calculated emissivities are therefore often not

representative of the actual emissivity of the urban environment. Why not use MODIS instead? It also has its limitations but has a more recurrent overpass. Lastly, I don't think that quitting the AHE for the evaluation is justified. I grasp your concern and I support it. But to me, since we don't know how it influences the LST, it is best to keep them. After all, the model is supposed to accurately represent urban LST with human influence, no?

- Lines 271 - 274: I don't understand the Evp150 case. Why are you changing the values because of a measurement at the IAP site? You force your model with the airport AWS right? Please clarify the reasoning here.

- Lines 287 - 290: Here I still did not understand if your model simulated the temperature at the rural site ? If not, you cannot compare UHIIs. In general I would advise to do the evaluation for 2m air temperature at both the urban and the rural site and not for UHIIs. This will be more indicative of where the model is having troubles representing the local climate.

- Lines 335 - 339: I have the feeling that you put a lot of trust in your model. I agree with your arguments but it may simply be related to how the model works, no?

- Lines 367 - 368: In my humble opinion, LSTs are only indicative of places that may be correlated with higher urban heat. But other factors play an important role and this is why I would be really cautious with the interpretations that come from one satellite image only.

- Section 3.2: Following the previous comment I would recommend to entirely rework this section. You draw a lot of conclusions on one satellite image only and you tend to expect a high level of correlation between the urban LST at 11 AM and the modelled air temperature. You would need to demonstrate this correlation with AWS observations in my opinion. Other possibility may be to use more satellite images to try to show the recurrent correlations. Otherwise, I would suggest to remove this part.

- Section 3.3: I don't see the added value of this section. Do we really learn something about the heterogeneity of the urban climate in Beijing here? Why did you focus specifically to the surroundings of the airport? You have an LCZ map. Why don't you compare the temperature distributions between LCZs for example?

- I really liked reading the section 3.4. It is highly indicative of potential deficiencies in the modelled climate during a heat wave events. That is, I believe, a strong part of the manuscript as heatwaves are of key interest for public health.

- Lines 485 - 489: Again, are you sure that AHE are the only explaining factor?

- Lines 512 - 517: Does this manuscript really support this argument? In the end, I felt that the manuscript was rather descriptive and did not have a general story line. I would recommend suppressing these lines as you do not test what urban planning solution may be more appropriate or not.

- Please revise all your figures and their captions so that they are easily understood by readers as stand-alone pieces. In particular, avoid using acronyms or explain them in the caption.

Minor Comments

- Lines 49 - 51: This sentence should be put in the first paragraphs as it is the increasing amount of population living in cities that drives urbanization and its related land-use land-cover (LULC) changes

- Line 57 : "estimated" instead of "estimates"

- Lines 66 - 69: Could you comment upon that result and their methodology ? In general, shouldn't we use multiple rural and urban sites to quantify the UHII? I know you introduce the concept just above but try to make it clearer why dense urban, suburban and rural meteorological stations' networks are required.

- Line 70: Add "surface" before "temperature"

- Line 73: What do you mean by "control variables"?

- Line 79: What about the viewing angles? Would you propose a range of optimal viewing angles?

- Line 80: The link between the two paragraphs needs to be improved. Urban climate models are not introduced previously and come a bit out of the blue.

- Line 150: What is the Normalized Building volume? How is it normalized? Please provide this information to the reader.

- Lines 195 - 197: I don't understand this sentence. OSM offers albedo values?

- Lines 235 - 239: Do the AHE still contribute to the heating of the air in the model? Please clarify.

- You have two Sections 2.5.

- Line 284: Please don't call it an UHII. It is only based on two automatic weather stations. You can talk about a difference between an urban and a rural station.

- Lines 294 - 299: Could be put in the discussion instead.

- Lines 302 - 305: This discussion does not seem to have its place here. It is not a pure evaluation but rather an additional perspective on the outcome of the study. Also, local characteristics may indeed be a factor but cannot justify as a whole the observed biases.

- Lines 357 - 358: Are SW radiation the only explaining factor or could winter AHE due to heating be also the cause of such a difference?

- Lines 355 - 361: Could this part go in the discussion?

---

## Referee Comment (RC2) · Anonymous Referee #3 · 6 Jan 2021

Review of Biggart et al

The manuscript investigates the ability of the urban climate component of the ADMS-Urban model to simulate the Urban Heat Island (UHI) in Beijing in summer and winter by comparison to temperature observations made at an urban and rural site and also to satellite-derived land surface temperatures. Different model simulations performed demonstrate the impact of anthropogenic heat emissions (AHE) and surface moisture levels. The base model underestimates the Urban Heat Island Increment (UHII) through the night in both seasons, but overestimates the daytime UHII in summer particularly during heatwaves. The nighttime modelled UHII could be increased

by enhancing the AHE suggesting that hotspots associated with dense inner-city road networks and building developments may be underestimated by the model at its current resolution. During the summer, in the daytime, the modelled UHII could be decreased by reducing the modelled surface resistance to evaporation. However, the blanket increase in urban moisture was found to reduce the correlation with satellite land surface temperatures suggesting that it is unresolved fine-scale green spaces at the urban site which influence near-surface temperatures in the daytime in the summer.

The authors recommend strategies aimed at reducing the daytime storage heat flux to decrease nighttime UHIIs in summer by reducing nocturnal heat release, hence lowering the cooling energy demand at night and therefore the contribution from AHEs to urban warming as well as urban planning strategies aimed at increasing the density of cooler spaces associated with green spaces and waterways.

The manuscript is generally well written and logically presented. I would value the authors addressing the following specific comments:

Line 102 – 104: It would be informative to add some discussion (either to the conclusions or at the end of section 3.1) on how the inclusion of AHEs and surface moisture to other cities where ADMS-Urban has been used (e.g. in Kuala Lumpur and Dublin) would likely impact the modelled UHI – can these changes improve model simulations in other cities?

Line 226 – 228: 'The nocturnal contribution from transportation is increased, following Biggart et al. (2020), to account for the influx of heavy-duty diesel trucks (HDDT) into urban Beijing after the daytime ban within the Fourth Ring Road (Zhang et al. 2019).' How significant is this influx of HDDT to the nocturnal AHE? Could the authors expand on the discussion on the impact increasing this contribution has to the modelled UHI?

Line 305: The authors are comparing model simulations to observations made at a different location in Beijing. Is there any reason why there may be strong local AHE at the IAP site relative to the other sites in Beijing where UHIs have been reported?

Line 312: 'Cao et al. (2016) found a strong correlation between high concentrations of particulate matter over urban areas and the nocturnal UHI for several megacities in China ...'– was any correlation seen between the measured model UHII discrepancy with observed PM loading, particularly in winter?

---

## Author Comment (AC1) · 31 Mar 2021

The comment was uploaded in the form of a supplement:
https://acp.copernicus.org/preprints/acp-2020-931/acp-2020-931-AC1-supplement.pdf

---

## Author Response (AR1)

We thank both reviewers for their detailed and insightful feedback on this study which has considerably improved the manuscript. Responses to each comment are structured as follows: (a) reviewer comment (in bold), (b) our response to the comment, (c) changes to the manuscript (in quotation marks and italics). In the revised manuscript modified text is highlighted using Track Changes.

**Referee #1**

**General Comments**

**The paper entitled "Modelling spatiotemporal variations of the canopy layer urban heat island in Beijing at the neighbourhood-scale" uses the ADMS-Urban climate model to depict the influence of the Beijing metropolitan area on the climate. Although this paper could be of substantial added value for the community, in particular by highlighting the strength of the ADMS-Urban model in comparison to other models, or also by deeply studying the biases during heat wave conditions, the paper sticks to a rather descriptive tone that does not seem to answer any research question. I have serious concerns about some methodological aspects that are given below as major comments. In particular the drawing of some conclusions based on one satellite image only cannot be considered as a strong evidence to sustain some arguments given by the authors. I would recommend improving the introduction to define the key research questions that will be answered in the paper. The methodological section will also need to be reworked to be more easily understood. In the results, there are numerous parts that could be implemented in the discussion part, depending on the research questions to be answered. Finally, I would like the authors to clearly enumerate the key outcomes of their research in the conclusions, their limitations, and potential guidelines to continue their effort.**

We thank the reviewer for their constructive feedback.

**Major Comments**

1. **Lines 70-79: Could you add something about the temporal coverage of satellites? Afterall, most of them only take snapshots of the SUHII and this further limits their use to understand when people may be at higher risks during the day.**

We thank the reviewer for this comment. Yes, high spatially resolved satellite data (e.g. Landsat 8) required for detailed land cover and temperature studies across urban areas generally has poor temporal resolution, limiting assessments of evolving heat risks throughout the day. Also, LSTs from satellites such as MODIS with higher temporal resolution have coarse spatial resolution. Page 2 Line 81 of the updated manuscript:

*"Furthermore, the use of high spatially resolved (~ 100 m) satellite data (e.g. Landsat 8) to understand diurnal heat risk variability across cities is often restricted by its poor temporal coverage; Landsat 8 LSTs are typically available every 16 days. Instruments such as the Moderate Resolution Imaging Spectroradiometer (MODIS) provide images four times a day (Hough et al. 2020) but have much coarser spatial resolution (~ 1000 m)."*

2. **Lines 103-104: There are many other works which implemented LCZ in regional climate models to study the UHI. Actually, the work from Alexander et al. (2015) used the SUEWS model that you describe as an urban energy balance model, while your model is an urban climate model. Also, I do not think that Alexander et al. (2015) used OSM for making their LCZ map. Please clarify these points here to make clearer what your study "builds upon".**

We thank the reviewer for this comment. We have added references to further urban climate modelling studies which implement LCZs on Page 3 and Line 118 of the manuscript. Also, the reviewer is correct that Alexander et al. (2015) did not combine OSM and LCZ data for simulations with the SUEWS model – the intention of lines 103-104 was to state that we advance previous use of LCZ data for UHI modelling studies by augmenting LCZs with OSM land use information. This has been clarified on Page 3 and Line 117 of the manuscript:

*"Several previous studies have implemented LCZs for UHI simulations, including in Dublin (Alexander et al. 2015), Madrid (Brousse et al. 2019) and Singapore (Mughal et al. 2019). We build on this earlier work by combining a LCZ map for Beijing with the locations of fine-scale green spaces, waterways and buildings from OpenStreetMap (OSM)."*

   3. **At the end of the Introduction I am missing some key research questions which you will try to answer. Your description on the challenges related to the use of remote sensing and in-situ measurements are properly highlighted in a very clear and efficient manner as well as the use of UCMs for coping with these limitations. But I don't really get what your study will bring further. Could you add a paragraph on that before detailing the contents of the subsequent section? Also, could you explain why you chose the ADMS-Urban model in particular? This could be explained in Section 2.1 too.**

We thank the reviewer for this comment. We have now clearly defined the key research objectives to be addressed in the paper, including the motivation for using the ADMS-Urban Temperature and Humidity model, before describing the structure and contents of the following sections. The manuscript has been updated on Page 3 from Line 103:

*"Here we incorporate LCZ data in neighbourhood-scale resolution (~100 m) urban climate simulations across Beijing using the ADMS-Urban Temperature and Humidity model (hereafter ADMS-Urban). This study aims to (a) quantify the relative impacts of urban surface properties and AHEs on Beijing's canopy layer UHI during winter and summer periods, (b) produce neighbourhood-scale spatial distributions of near-surface air temperatures across urban Beijing and explore how they vary diurnally in summer, and (c) understand the extent to which summer heatwave periods affect daytime and nighttime UHIs in Beijing."*

*ADMS-Urban is chosen for this work given it can be used to capture the impact of fine-scale land cover variations on urban climate, therefore highlighting the microclimates where residents are most at risk from extreme temperatures and informing urban planners on the cooling effects of green spaces and waterways. This local-scale urban climate model…"*

   4. **Line 127: What do you define as "upwind"? How do you get this information since it is at the baseline of your model from what I understand. I am not really familiar with the ADMS-Urban climate model and I miss a bit how it actually integrates information at the boundaries to the inner parts of the domain.**

We appreciate the reviewer's query about the definition of upwind conditions and have attempted to make this clearer. The manuscript has been updated on Page 4 from Line 128:

*"…ADMS-Urban climate model calculates local perturbations to vertical profiles of temperature and humidity, representative of rural conditions upwind of the modelled urban area, in response to spatially varying neighbourhood-scale surface parameters. These upwind profiles are calculated from near-surface meteorological measurements recorded outside the modelled urban area, ideally coinciding with air advecting towards the urban centre (i.e. upwind), and depend on the planetary boundary layer*

height (PBLH), surface roughness length ($z_0$) and the stability parameter $PBLH/L_{MO}$. $L_{MO}$ is the Monin-Obukhov length…"

Upwind surface net radiation (Q*), which governs the modelled upwind surface heat fluxes, is determined from upwind near-surface air temperature measurements and surface albedo values defined at the edge of the modelled urban area, which represent conditions at the upwind meteorological site. This is clarified on Page 4 from Line 148:

"…which is dependent on the upwind surface albedo (r) defined around the edge of the modelled urban area."

Calculations of local temperature and humidity perturbations due to local surface heat fluxes and surface roughness values are based on differences between 'upwind' surface properties defined around the edge of the model domain and the spatially varying surface properties defined across the urban area. This is clarified on Page 4 Line 148:

"Local perturbations (Δ) to the upwind surface heat fluxes are calculated based on relative differences between upwind surface parameters defined around the edge of the modelled urban area and the spatially varying surface characteristics defined across the model domain, according to the surface energy balance equation:"

The upwind-urban surface parameter differences governing the perturbations to each surface heat flux term are clearly defined on Page 5 from Line 172-177.

**5.1 Lines 160-174: Here you say the model is driven by rural stations. Are they at the boundary of your domain? You use only the Airport weather station to force your model what I understand. What are the potential limitations to that for an extended city as Beijing? Did you also consider that the airport is located in a rather built up environment (when looking at your Fig.1)? Why not using the Pinggu Station instead?**

We thank the reviewer for this comment. There seems to be some confusion concerning the meteorological data used for this study. This may, in part, be due to referring to both 'rural' and 'upwind' meteorology separately. We have removed:

"Rural meteorological data for driving the model are available for the duration of both campaigns"

And have updated the manuscript on Page 5 Line 183:

"For this study, a single set of meteorological measurements, representing upwind conditions, are used to drive the model."

As stated on Lines 184-187, the meteorological data used to drive the model comes from both the Pinggu station (air temperature and relative humidity) and the airport (wind speed, wind direction and cloud cover). On Line X we explain that airport wind speed and wind direction data are used instead of the measurements at Pinggu as the airport is in a more exposed location and the airport site is subject to WMO quality assurance – thus, overall we believe the airport wind speed/direction measurements are more representative of the full urban Beijing area than those from Pinggu. The manuscript is updated on Line 190:

"Measurements of wind speed and direction from the airport site are used as the 'rural', or upwind data, instead of those from the rural site, as it is subject to WMO quality assurance and is likely to be less influenced by local frictional effects of neighbouring small buildings and vegetation, compared to the rural site, due its more exposed location."

The reviewer is correct in highlighting the limitations of using a single set of meteorological measurements to represent conditions across an urban area as large as Beijing. But, of course, the purpose of the model is to perturb upwind meteorology locally; the role of spatially varying surface properties in perturbing surface heat fluxes, and therefore air temperature and humidity, has been previously explained. The combined effect of local heat flux perturbations and spatially varying surface roughness on PBL stability also modifies vertical wind speed profiles across the domain. The manuscript is updated on Page 5 Line 193:

*"Upwind vertical wind speed profiles are perturbed locally in the model, impacting heat advection, following modifications to PBL stability due to spatially varying heat fluxes and surface roughness."*

**5.2 Also, do we have to understand that the winter and the summer domains are located at the same place based on the winter wind direction? In general I had a hard time understanding which data is used to drive the model, what is your domain extension, its horizontal resolution and why you chose those three stations among others.**

On Page 5 Line 198-201 we explain the limitation of only using a single upwind meteorological site for air temperature measurements; when the wind direction is not from the northeast, the modelled urban areas is not situated downwind of the temperature measurement. Ideally, we would use several upwind sites located around the model domain, selecting the air temperature coinciding with upwind conditions. The manuscript is updated on Page 5 Line 195:

*"Incorporating near-surface air temperature measurements from a single rural meteorological station is a limitation of this work; ideally, we would use several rural meteorological sites distributed around the model domain, selecting the air temperature measurement from the site coinciding with upwind conditions. For this study, the only appropriate upwind air temperature measurement data available was from the rural field campaign site at Pinggu. However, during the winter period…"*

The domain extent and horizontal resolution are detailed later in Sect. 2.3.

**6. Lines 191-195: How did you get these values of 200 and 150? What is the rationale for saying that one is representative of densely built-up LCZ and the other of more open urban typologies? Please explain.**

We thank the reviewer for this comment. A surface resistance to evaporation value of 200 s m$^{-1}$ is widely reported in the literature (see manuscript for references) to characterise urban surface materials. However, the high density of green spaces and the prevalence of road wetting for cleaning purposes across urban Beijing, noted by Dou et al. (2019), is likely to lower the surface resistance to evaporation, particularly across LCZ categories described as comprising abundant pervious land cover. Given the strong influence on daytime temperatures of surface resistance to evaporation (especially in summer), we reduce values to 150 s m$^{-1}$ in LCZ 4-6 and 9, with the aim of increasing the spatial heterogeneity of modelled air temperatures across urban Beijing. The value of 150 s m$^{-1}$ is chosen as it is an approximate mid-point between the surface resistance to evaporation allocated to green spaces/forests and urban surfaces. To make this reasoning clearer in the manuscript, we update from Page 6 Line 222:

*"A surface resistance to evaporation of 200 s m$^{-1}$ is widely deemed in the literature to characterise urban surfaces, thus we assign it to LCZs described by Stewart and Oke (2012) as consisting entirely of impervious materials (LCZ 1-3, 8 and 10). However, to account for the high density of green spaces and prevalence of road wetting for cleaning purposes across urban Beijing, noted by Dou et al. (2019), we lower the surface resistance to evaporation assigned to LCZs described by Stewart and Oke (2012) as*

*consisting of abundant pervious land cover (e.g. plants and trees) (LCZ 4-6 and 9) to 150 s m$^{-1}$; this value is an approximate midpoint between the surface resistance to evaporation values given to green spaces and urban surfaces (Table A1 and Table 1). Given the strong influence on daytime air temperatures of surface resistance to evaporation, this adjustment is made to ensure sufficient spatial heterogeneity in modelled air temperature is captured across central urban areas."*

7. **Lines 200-211: This paragraph partially answers some of my previous comments but I still have a very hard time understanding what is "upwind". If it is a border surrounding a domain then it means that all winds converge to the city center? In general I think that the methodology part will need a complete restructuring.**

We thank the reviewer for this comment. To clarify, when we say 'model domain' in the manuscript, this is referring to the area within the Sixth Ring Road across which air temperatures are modelled. Surface properties (e.g. thermal admittance, surface resistance to evaporation etc) at the upwind meteorological site are represented by a 1 km border around the perimeter of each map in Fig. 3. The surface information between the modelled area (Sixth Ring Road) and the 1 km border in Fig. 3 is a 'buffer zone' required in the model to prevent erroneous results associated with a sudden transition between urban and upwind surface properties. Modelled air temperature perturbations across the area within the Sixth Ring Road are based on differences between local and upwind surface properties.

We have made the following updates to the manuscript to clarify the above. On Page 6 Line 207:

*"Near-surface air temperatures are modelled across the area contained within Beijing's Sixth Ring Road (Fig. 1); the resolution of the model calculation grid is ~ 105 m. Thermal and morphological properties covering the modelled urban area and surrounding suburban regions are derived from OpenStreetMap…"*

On Page 6 Line 23:

*"Surface characteristics at the upwind meteorological site are represented by a 1 km border extending around the perimeter of each thermal and morphological surface parameter map in Fig. 3. Differences between upwind surface parameters and those within the model domain are used to calculate the urban temperature perturbations (Sect. 2.1). The surface information defined between the Sixth Ring Road and the 1 km upwind border (Fig. 3), covering suburban areas, is required in the model to prevent erroneous simulated temperatures associated with a sharp transition between urban and rural surface parameters."*

8. **Line 219-221: Please explain how this is spatialized throughout the domain and provide the reader with an equation on the scaling from one LCZ to another.**

We thank the reviewer for this comment. First, the summer (June, July, August) AHE value estimated for the IAP site by Dou et al. (2019) is scaled to an annual mean AHE value based on the monthly scaling factors from Lu et al. (2016) (Fig. 5). As IAP is located in LCZ 1, this annual mean AHE for IAP is distributed spatially by multiplying by the AHE magnitude estimated for each LCZ class by Stewart and Oke (2012) relative to the AHE value (mid-point of suggested range) estimated for LCZ 1. The manuscript is updated on Page 7 Line 257:

*"First, the IAP value for summer (mean for June, July and August) is scaled to represent an annual mean AHE value based on literature-reported monthly scaling factors (Fig. 5) (Lu et al. 2016). This annual mean AHE for IAP is distributed spatially through further scaling by the magnitudes of AHE estimated for each LCZ class by Stewart and Oke (2012) relative to to the AHE value for LCZ 1 (IAP is situated in LCZ 1), according to Eq. (4):*

$$AHE_i = \frac{X_i}{X_1} \times \frac{Y_{JJA}}{MSF_{JJA}}$$ (4)

where, $AHE_i$ is the annual mean AHE value for LCZ $i$ ($i$ = 11-15 for LCZ A, B, D, E and G, respectively). $X_i$ and $X_1$ represent the AHE value estimated by Stewart and Oke (2012) for LCZ $i$ and 1 (midpoint of range), respectively. $Y_{JJA}$ is the summer (June, July and August) mean AHE calculated for IAP by Dou et al. (2019) and $MSF_{JJA}$ gives the summer mean monthly scaling factor (Fig. 5)."

9. **Section 2.5 is among my main concerns in the methodology. I somewhat understand the reasoning behind those choices, but I don't think that using LST to evaluate air temperature is a good thing to do. Additionally, the use of only one Landsat 8 image for evaluating a model that is run for 2 months seems really limited to me. To that I would like to add that Landsat 8 LSTs retrieved by Jimenez and Munoz techniques are directly correlated to the NDVI. The calculated emissivities are therefore often not representative of the actual emissivity of the urban environment. Why not use MODIS instead? It also has is limitations but has a more recurrent overpass. Lastly, I don't think quitting the AHE for the evaluation is justified. I grasp your concern and I support it. But to me, since we don't know how it influences the LST, it is best to keep them. After all, the model is supposed to accurately represent urban LST with human influence, no?**

We thank the reviewer for highlighting their concerns with Sect. 2.5.

For this study, in-situ air temperature measurements in urban Beijing (other than at the field campaign site at IAP) were not available; ideally, a dense network of urban weather stations would be used to evaluate spatial variations in modelled near-surface air temperatures. Therefore, we decided to use satellite-derived LSTs as a means of providing some form of evaluation of modelled air temperature spatial variability.

This technique was also adopted by Wang et al. (2019) when using ADMS-Urban to model the UHI in Kuala Lumpur. The manuscript is updated on Page 8 Line 307:

*"This technique was also adopted by K. Wang et al. (2019) for high-resolution UHI simulations across Kuala Lumpur using ADMS-Urban to assist in the evaluation of modelled near-surface air temperature spatial patterns."*

Given the strong incoming SW radiation in summer in Beijing, we expect the air temperature at 2.5 m, which is heavily influenced by surface heat fluxes, to be reasonably coupled with the LST. We highlight the expected differences between near-surface air temperatures in Sect. 2.1, and throughout Sect. 3.2, due primarily to micro-scale advection of heat; thus we acknowledge that a strong correlation between LSTs and modelled 2.5 m air temperature is not expected. Furthermore, by increasing urban surface moisture (Evp150 case), modelled daytime air temperature more closely agrees with measurements at IAP but becomes more uncoupled with the LSTs, therefore highlighting the expected differences between the two properties, which we believe is a useful outcome from this study.

We appreciate the concerns raised regarding the use of a single Landsat 8 satellite image. As LSTs derived from Landsat 8 data are available every 16 days, there was only 1 other image available during the summer campaign period, on 8 June; however, there was significant cloud cover over Beijing on this day, thus the satellite image could not be used to accurately derive LSTs. MODIS-derived LSTs are available 4 times per day, but at a much coarser spatial resolution (1000 m), thus could not be used to evaluate the neighbourhood-scale (~ 100 m) spatial variability of modelled 2.5 m air temperature. Specifically, the ability of the model to accurately capture the impact of fine-scale land cover features (e.g. green spaces and waterways) on near-surface air temperature could not be evaluated using

MODIS satellite data. Overall, we believe the single Landsat 8 satellite image used here provides a useful guide for evaluating modelled spatial temperature variability and adds sufficient value to remain in the manuscript.

Finally, we feel the exclusion of AHEs in simulations of near-surface air temperature used for comparison with LSTs is justified. From results presented and discussed in Sect. 3.1 we know that AHEs dominate modelled UHIIs. We also know that, in general, the micro-advection of heat released by the surface strongly impacts air temperatures and contributes to their uncoupling with LSTs. Therefore, it is likely that the horizontal advection of anthropogenic heat has a more significant effect on air temperatures than LSTs, increasing the uncoupling between air and surface temperatures and further limiting the value of their comparison. This would also make it more difficult to evaluate the success of combining high-resolution land cover information for UHI simulations. The manuscript has been updated on Page 9 Line 316:

*"LSTs are compared with simulations that exclude AHEs as (a) previous studies have reported difficulties in determining the impact of AHEs on LSTs (Kato and Yamaguchi, 2005; Wang et al. 2017), and (b) micro-scale advection of heat released by nearby surfaces is known to uncouple LSTs and air temperatures (Roth et al. 1989; Voogt and Oke, 1998), thus we expect the release of anthropogenic heat to have a similarly strong influence on near-surface air temperatures and further contribute to differences with spatial LST variability."*

10. **Lines 271-274: I don't understand the Evp150 case. Why are you changing the values because of a measurement at the IAP site? You force your model with the airport AWS right? Please clarify the reasoning here.**

We thank the reviewer for this comment. As stated on Page 5 Line 184, we drive the model with air temperature measurements from the rural site at Pinggu (we use wind speed and direction from the airport AWS). The model is used to calculate air temperature changes at the IAP site, relative to the rural measurements, due to differences between upwind (Pinggu) and local (IAP) surface thermal and morphological parameters. Unusually high latent heat flux values have previously been observed at IAP (Dou et al. 2019), hence for the evp150 case we lower the urban surface resistance to evaporation with the aim of improving the agreement between measured and modelled UHIIs at IAP. We explain the general aim of all model experiments on Page 9 Line 324.

11. **Lines 287-290: Here I still did not understand if your model simulated the temperature at the rural site? If not, you cannot compare UHIIs. In general I would advise to do the evaluation for 2 m air temperature at both the urban and the rural site and not for UHIIs. This will be more indicative of where the model is having troubles representing the local climate.**

The reviewer's comment here seems to be a continuation of a misunderstanding regarding which air temperature measurements are used in the model. The UHII is defined as the difference in measured air temperatures at the IAP site (urban) and the Pinggu site (rural). The model is used to calculate the local perturbation at IAP to upwind temperature profiles derived from the Pinggu air temperature measurements (i.e. the UHII). We believe this is made clear in the manuscript.

12. **Lines 335 - 339: I have the feeling that you put a lot of trust in your model. I agree with your arguments but it may simply be related to how the model works, no?**

We thank the reviewer for their comment. We feel that the arguments made to account for modelled daytime UHII overestimations are valid. Underestimations of afternoon heat storage correlate well

with modelled underestimations of nocturnal UHIIs (insufficient evening heat release in the model), which are enhanced during heatwaves. Furthermore, the effects of enhanced evaporative cooling are tested with the Evp150 case and substantially improve daytime measured-modelled UHII agreement.

**13. Lines 367-368: In my humble opinion, LSTs are only indicative of places that may be correlated with higher urban heat. But other factors play an important role and this is why I would be really cautious with the interpretations that come from one satellite image only.**

The lines referred to here by the reviewer discuss the information that can be gained from high-resolution maps of near-surface air temperature, not LSTs.

**14. Section 3.2: Following the previous comment I would recommend to entirely rework this section. You draw a lot of conclusions on one satellite image only and you tend to expect a high level of correlation between the urban LST at 11 AM and the modelled air temperature. You would need to demonstrate this correlation with AWS observations in my opinion. Other possibility may be to use more satellite images to try to show the recurrent correlations. Otherwise, I would suggest to remove this part.**

We thank the reviewer for this comment. As previously stated in response to Comment #9, access to additional air temperature measurements at other urban meteorological stations was not possible for this study; thus, we could not evaluate the spatial variability of modelled air temperatures following the method suggested here by the reviewer. Furthermore, as also explained in response to Comment #9, high-resolution Landsat 8 satellite images are available every 16 days, therefore we are unable to test the recurrency of the LST-air temperature correlations during the summer campaign period. We have updated the manuscript on Page 11 Line 430 to acknowledge this study limitation:

*"Due to the infrequency of Landsat 8 satellite image availability (every 16 days), we are unable to test how variable the spatial correlation between LSTs and modelled near-surface air temperatures is with time. However, the comparison for this single hour during the summer period provides a useful guide for assessing the general model perfromance in capturing urban Beijing's neighbourhood-scale air temperature patterns using a hybrid of LCZ and OSM land cover data."*

We clearly reiterate the limitations associated with comparing LSTs and air temperatures on Page 12 Line 437. We update the manuscript on Page 12 Line 445 to emphasise that strong spatial correlations are not expected:

"These inherent differences between the two variables limit direct comparisons and thus the strength of the correlation to be expected; however, the relative spatial patterns are of interest."

We previously attempted to explain how LST-air temperature spatial agreement could be increased, "may require more detailed sub-divisions of urban land cover…". This sentence has now been removed as increasing the correlation is likely both not possible and not desirable given the inherent differences between LSTs and air temperature.

The reviewer remarks that we draw a lot of conclusions from one satellite image. However, in Sect. 3.2 there are only two main conclusions made:

1. Increasing urban moisture levels (Evp150 case) improves agreement between daytime measured and modelled air temperatures at IAP but lowers the spatial correlation with LSTs. This outcome highlights the expected differences between LSTs and air temperatures – the numerous reasons for their possible uncoupling have been explained previously (e.g. micro-advection and turbulent mixing of thermally contrasting pockets of air)

2. Urban cool islands associated with green spaces and waterways are successfully captured by the model and the land use data implemented for this study. The lowest LSTs are more likely to be spatially correlated with the lowest nears-surface air temperatures as you would expect there to be less turbulent mixing over these cooler regions.

**15. Section 3.3: I don't see the added value of this section. Do we really learn something about the heterogeneity of the urban climate in Beijing here? Why did you focus specifically to the surroundings of the airport? You have an LCZ map. Why don't you compare the temperature distributions between LCZs for example?**

We thank the reviewer for this comment. The aim of Sect. 3.3 was to build on the air temperature distributions presented in Sect. 3.2 for 11 am, illustrating the extent to which the heterogeneity of Beijing's urban climate varies diurnally. We focus specifically on the airport and its surroundings to demonstrate the impact that large-scale urban developments can have on local climate, a similar approach to that adopted by Hamilton et al. (2014) when investigating the impact of the Olympic Park development on local climate in London. Diurnal variability in the differences between air temperatures at the airport and nearby forest and river locations is quantified, utilising the fine-scale land use information provided by OpenStreetMap. Overall, we believe this section provides useful information for urban planners on the potential effects of large urban developments on human thermal comfort, cooling energy demand and air quality, in addition to the extreme temperature mitigation effects that vegetative surfaces and waterways can have and therefore remains unchanged in the manuscript.

Air temperature comparisons between LCZs is an interesting suggestion for future work and may provide urban planners with more general guidance regarding the climatic impact of urban structure and form, as opposed to the impact of specific urban developments.

**16. Lines 485-489: Again, are you sure that AHE are the only explaining factor?**

Underestimated nocturnal UHIIs in winter are most likely related to low AHEs used in the model, yes. It is possible that, similar to the summer, the model slightly underestimates the magnitude of stored heat released at night; however, given the comparatively weak incoming solar radiation in winter, this effect is likely small. In Sect. 3.1 we add that a previous study by Cao et al. (2016) found a strong correlation between particulate matter concentrations at night and the UHI magnitude in China related to the emission of LW radiation towards the surface at night by haze pollution; as the model does not account for LW radiation emission by aerosols, it is possible that this further contributes to nocturnal UHII underestimations. However, in Sect. 4, the paper's primary findings are outlined, and we feel low AHEs are the most substantial cause of low modelled UHIIs in winter.

**17. Lines 512-517: Does this manuscript really support this argument? In the end, I felt that the manuscript was rather descriptive and did not have a general story line. I would recommend supressing these lines as you do not test what urban planning solution may be more appropriate or not.**

We thank the reviewer for this comment. It is true that the impact on air temperatures of specific UHI mitigation techniques is not tested in this study. However, through the various sensitivity simulations and spatial air temperature distributions presented and discussed, we clearly highlight the dominant mechanisms driving Beijing's UHI (e.g. AHEs or ground heat storage and release) and the cooling

influence of green spaces and waterways; thus provide critical information for urban planners when considering the impact on local climate of future developments.

18. **Please revise all your figures and their captions so that they are easily understood by readers as stand-alone pieces. In particular, avoid using acronyms or explain them in the caption.**

We thank the reviewer for this comment. All sources have data and acronyms (e.g. LCZs), apart from the model experiment names, were previously defined in the figure captions. We have updated the manuscript so that model experiment names included in figure captions are described:

*"Figure 6. Mean diurnal variation in measured and modelled urban heat island intensities (UHIIs) at the urban site in (a) winter and (b) summer. Modelled UHIIs from Base (black dotted), anthropogenic heat emissions + 50 % (AHE50; black dashed), high urban moisture (Evp150; blue) and anthropogenic heat emissions excluded (noAHE; pink) cases are presented. Measurements are marked by the red line. Shaded regions and error bars represent the 95 % confidence intervals for modelled and measured UHIIs, respectively."*

*"Figure 7. Hourly measured and modelled urban heat island intensities (UHIIs) for (a) winter and (b) summer periods. Modelled UHIIs from high urban moisture (Evp150; blue), anthropogenic heat emissions + 50 % (AHE50; grey) and anthropogenic heat emissions excluded (noAHE; pink) cases are presented. Measured UHIIs are grouped into bins (0.5 °C), points representing the mean modelled UHII in the bin. Point sizes scaled by total number of hourly values per bin. Error bars represent 1 standard deviation of hourly modelled UHIIs in each bin."*

*"Figure 8. Near-surface (2.5 m above ground level) air temperatures and land surface temperatures (LST) within Beijing's Sixth Ring Road on 23 May 2017. Air temperatures modelled at 11:00 am, excluding anthropogenic heat emissions, with surface parameters from (a) Base, and (b) high urban moisture (Evp150) cases. (c) Differences between air temperatures modelled in (a) and (b) (a minus b). (d) Landsat 8-derived LSTs (USGS, 2020) at 10:53 am. Beijing's Fifth (outer) and Third (inner) Ring Roads marked by white (a, b and d) and black (c) lines."*

*"Figure 9. Frequency (colour) of 100 m resolution Landsat 8-derived land surface temperatures (USGS, 2020) and near-surface (2.5 m above ground level) air temperatures, modelled with surface parameters from (a) Base, and (b) high urban moisture (Evp150) cases, both excluding anthropogenic heat emissions."*

*"Figure 14. Mean measured (red) and high urban moisture (Evp150) case modelled (blue) urban heat island intensities (UHIIs) for heatwave (HW) and non-heatwave (non-HW) days (Fig. 13) for day (10:00 to 16:00) and night (22:00 to 04:00) hours. Error bars represent 1 standard deviation of daily mean day and night UHIIs. Mean UHIIs for each period are marked either above or below error bars."*

**Minor Comments**

19. **Lines 49 - 51: This sentence should be put in the first paragraphs as it is the increasing amount of population living in cities that drives urbanization and its related land-use land-cover (LULC) changes.**

We thank the reviewer for this comment, however the lines referred to here are placed at the beginning of the second paragraph as it specifically describes the extent of urbanisation in China (followed by relevant information on heatwaves in China and their ageing population). The first paragraph contains a general definition of the UHI and a description of the key driving mechanisms.

Furthermore, the increase of urban populations is part of the definition of urbanisation rather than a driving mechanism. The manuscript remains unchanged.

**20. Line 57: "estimated" instead of "estimates"**

The manuscript has been changed as suggested. (Page 2 Line 57)

**21. Lines 66 - 69: Could you comment upon that result and their methodology? In general, shouldn't we use multiple rural and urban sites to quantify the UHII? I know you introduce the concept just above but try to make it clearer why dense urban, suburban and rural meteorological stations' networks are required.**

We thank the reviewer for this comment. For the Jiang et al. (2019) study, UHIIs were determined as the difference between urban air temperature measurements and the average air temperature measured across several rural stations. When rural stations situated near the coast were used instead of inland rural stations, the daily maximum UHII would occur during the daytime, rather than at night, due to the cooling effect of the sea breeze during the afternoon. The manuscript is updated on Page 2 Line 68:

"…depending on whether inland or coastal rural reference sites were chosen, due to the cooling effects of the daytime sea breeze."

**22. Line 70: Add "surface" before "temperature"**

The manuscript has been changed as suggested. (Page 2 Line 70)

**23. Line 79: What about the viewing angles? Would you propose a range of optimal viewing angles?**

We thank the reviewer for this comment. At elevation angles approaching 90° rooftops can become oversampled across urban areas. Rooftops are generally constructed from low albedo and low thermal admittance materials and therefore readily absorb and release solar radiation, leading to anomalously large diurnal ranges in satellite-derived LSTs. Elevation angles can be as low as 30°, increasing the proportion of vertical surfaces (i.e. building walls) viewed, which generally have higher thermal inertia than roofs. Satellite image pixel values represent the average radiances from all surfaces within the pixel area. Therefore, optimal satellite viewing angles should capture representative samples from horizontal and vertical surfaces. The manuscript is updated on Page 2 Line 77:

"….walls or roofs of buildings may become oversampled at particularly low (~ 30 °) or high (~ 90 °) satellite viewing angles, with roofs typically constructed from lower thermal inertia materials than walls generating greater diurnal LST variability…"

**24. Line 80: The link between the two paragraphs needs to be improved. Urban climate models are not introduced previously and come a bit out of the blue.**

We thank the reviewer for this comment. We have updated the manuscript on Page 3 from Line 85 accordingly:

*"Urban climate models can be used to produce complete spatially and temporally resolved air temperature distributions across cities and thus provide a solution to the poor spatial coverage of air temperature measurements and the limitations associated with the derivation of LSTs from satellite data. Fine-scale modelling of urban climate is frequently undertaken…"*

**25. Line 150: What is the Normalized Building volume? How is it normalized? Please provide this information to the reader.**

The normalised building volume (NBV) is a measure of the density of the buildings within the domain. It is defined for each grid cell as the volume of buildings within a grid cell per grid cell surface area. The manuscript is updated on Page 5 Line 170:

*"NBV is defined as the volume of buildings within a grid cell per grid cell area and provides a measure of the density of buildings within the domain."*

**26. Lines 195-197: I don't understand this sentence. OSM offers albedo values?**

We thank the reviewer for this comment. The sentence referred to is explaining that, for the purposes of consistency in surface parameters between land use data sources, the albedo values assigned to LCZ classes correspond to the lower end of the ranges given by Stewart and Oke (2012) as these match closely with the literature-reported albedo values we have given to the OSM data.

**27. Lines 235-239: Do the AHE still contribute to the heating of the air in the model?**

Yes, the dispersion of AHEs (units of $W/m^2$) is modelled in the same way as air pollutant emissions with the ADMS-Urban air quality model (i.e. as Gaussian plumes). The accumulation of dispersed Gaussian plumes of heat energy gives the 'energy density field', with units of $J/m^3$. The energy density field ($C_T$) is converted to a temperature increment ($\Delta T$) according to:

$$\Delta T = \frac{C_T}{\rho c_p}$$

Where $\rho$ (1.225 $kg/m^3$) and $c_p$ (1012 kJ/kgK) are standard values for the density and specific heat capacity of air at 15 °C. The manuscript has been updated on Page 8 Line 285:

*"The accumulation of dispersed plumes of anthropogenic heat gives an energy density field ($C_T$), in units of $J\ m^{-3}$, which is converted to a local air temperature increment ($\Delta T$) in the model according to:*

$$\Delta T = \frac{C_T}{\rho c_p} \tag{X}$$

*where $\rho$ (1.225 $kg/m^3$) and $c_p$ (1012 kJ/kgK) are standard values for the density and specific heat capacity of air at 15 °C (CERC, 2018)."*

**28. You have two Sections 2.5.**

The manuscript has been updated as required.

**29. Please don't call it an UHII. It is only based on two automatic weather stations. You can talk about a difference between an urban and a rural station.**

We thank the reviewer for this comment. We agree that, strictly speaking, an urban heat island intensity (UHII) should refer to the difference between urban and rural temperature measurements averaged across multiple weather stations. However, differences between air temperatures measured at single urban and rural meteorological stations have previously been referred to as UHIIs (Barlow et al. 2015) and as we extensively use the term UHII throughout the study, at this stage we have to leave the manuscript unchanged. Also, referring to the UHII as an urban-rural difference throughout would greatly add to the wordiness of the manuscript.

**30. Lines 294-299: Could be put in the discussion instead.**

Section 3 combines both the Results and Discussion. The lines referred to by the reviewer are appropriate for paragraph two of Sect. 3.1 as they add context to the UHIIs measured for Beijing in this study and introduce the importance of mechanisms such as ground heat storage and release that likely contribute strongly to the measured-modelled nocturnal UHII differences in summer presented in the subsequent paragraphs. The manuscript remains unchanged.

**31. Lines 302 - 305: This discussion does not seem to have its place here. It is not a pure evaluation but rather an additional perspective on the outcome of the study. Also, local characteristics may indeed be a factor but cannot justify as a whole the observed biases.**

We thank the reviewer for this comment. The lines referred to by the reviewer are included to illustrate that the modelled nighttime UHIIs in this study agree closely with previous measurements at different sites in Beijing; this suggests that the model is successful in capturing the mean UHI magnitude in Beijing but that local site characteristics at IAP differentiating its UHII from other parts of the city are not represented in the model setup. The manuscript remains unchanged.

**32. Lines 347-348: Are SW radiation the only explaining factor or could winter AHE due to heating be also the cause of such a difference?**

It is not clear what the reviewer is querying here. In the lines referred to in the study we explain the differences between winter and summer modelled diurnal temperature ranges when AHEs are excluded from simulations.

**33. Lines 355-361: Could this part go in the discussion?**

Again, it is not clear what the reviewer is asking for here. Section 3 includes Results AND Discussion. The manuscript remains unchanged.

**Referee #2**

**Review of Biggart et al**

**The manuscript investigates the ability of the urban climate component of the ADMS-Urban model to simulate the Urban Heat Island (UHI) in Beijing in summer and winter by comparison to temperature observations made at an urban and rural site and also to satellite-derived land surface temperatures. Different model simulations performed demonstrate the impact of anthropogenic heat emissions (AHE) and surface moisture levels. The base model underestimates the Urban Heat Island Increment (UHII) through the night in both seasons, but overestimates the daytime UHII in summer particularly during heatwaves. The nighttime modelled UHII could be increased by enhancing the AHE suggesting that hotspots associated with dense inner-city road networks and building developments may be underestimated by the model at its current resolution. During the summer, in the daytime, the modelled UHII could be decreased by reducing the modelled surface resistance to evaporation. However, the blanket increase in urban moisture was found to reduce the correlation with satellite land surface temperatures suggesting that it is unresolved fine-scale green spaces at the urban site which influence near-surface temperatures in the daytime in the summer. The authors recommend strategies aimed at reducing the daytime storage heat flux to decrease nighttime UHIIs in summer by reducing nocturnal heat release, hence lowering the cooling energy demand at night and therefore the contribution from AHEs to urban warming as well as urban planning strategies aimed at increasing the density of cooler spaces associated with green spaces and waterways.**

We thank the reviewer for their positive feedback.

**The manuscript is generally well written and logically presented. I would value the authors addressing the following specific comments:**

1. **Line 102 – 104: It would be informative to add some discussion (either to the conclusions or at the end of section 3.1) on how the inclusion of AHEs and surface moisture to other cities where ADMS-Urban has been used (e.g. in Kuala Lumpur and London) would likely impact the modelled UHI – can these changes improve model simulations in other cities?**

We thank the reviewer for this comment. It is difficult to estimate the potential impact of including/increasing AHEs and increasing surface moisture on model perfromance for the previous applications of ADMS-Urban in Kuala Lumpur and London as neither study evaluates the model output against measurements at in-situ meteorological stations. For our study, increasing the magnitude of AHEs and enhancing the surface moisture enabled the model to better capture the local UHI characteristics at IAP. These adjustments provide guidance for possible ways to improve measured-modelled agreement in other cities but is recommended to begin with a base model set-up, as we demonstrated, and then test the sensitivity of modelled UHIIs to different surface parameters.

2. **Line 226 – 228: 'The nocturnal contribution from transportation is increased, following Biggart et al. (2020), to account for the influx of heavy-duty diesel trucks (HDDT) into urban Beijing after the daytime ban within the Fourth Ring Road (Zhang et al. 2019).' How significant is this influx of HDDT to the nocturnal AHE? Could the authors expand on the discussion on the impact increasing this contribution has to the modelled UHI?**

We thank the reviewer for this comment. For the Biggart et al. (2020) study, the proportion of air pollutant emissions released at night was increased approximately by a factor of 2 to account for the influx of HDDTs into central Beijing following the lifting of traffic restrictions from 23:00 to 06:00; this particularly improved the agreement between measured and modelled $NO_2$ concentrations at night. For this study, the diurnal AHE profile estimated by Lu et al. (2016) is similarly adjusted at night, assuming approximate proportionality between air pollutant emissions and AHEs. The manuscript is updated on Page 7 Line 273:

*"….after the daytime ban (06:00-23:00) within the Fourth Ring Road (Zhang et al. 2019). We assume approximate proportionality between air pollutant emissions (Biggart et al. 2020) and AHEs from HDDTs and increase the nocturnal transportation sector AHE component estimated by Lu et al. (2016) by ~ a factor of 2."*

3. **Line 305: The authors are comparing model simulations to observations made at a different location in Beijing. Is there any reason why there may be strong local AHE at the IAP site relative to the other sites in Beijing where UHIs have been reported?**

The IAP site is situated close to a busy road (~ 110 m) and is surrounded by high-rise buildings, as stated on Line X. It is difficult to comment on how this compares to sites used for other studies without detailed descriptions of their local site characteristics. However, two of the four urban meteorological stations used by Wang et al. (2017) in Beijing are described as being located in green belts/parks, which should be less influenced by local sources of AHEs. The manuscript is updated on Page 10 Line 369:

*"Two of the four urban meteorological stations in Beijing used by Wang et al. (2017) are situated in green belts or parks, thus are likely to be less influenced by major local sources of AHEs, which may account for the lower nocturnal UHIIs observed there relative to IAP."*

4.  **Line 312: 'Cao et al. (2016) found a strong correlation between high concentrations of particulate matter over urban areas and the nocturnal UHI for several megacities in China . . .'– was any correlation seen between the measured model UHII discrepancy with observed PM loading, particularly in winter?**

We thank the reviewer for this comment. We investigated this, however, $PM_{2.5}$ concentrations measured at Pinggu were often of a similar magnitude or even greater than those measured in urban Beijing during the winter period. The positive correlation observed by Cao et al. (2016) was based on an aerosol loading increment over urban areas in China relative to neighbouring rural areas, with resulting downwelling LW radiation enhancements over cities increasing the UHII. High $PM_{2.5}$ levels at Pinggu may be a result of the general regional-scale extent of haze pollution in northern China or strong local sources of $PM_{2.5}$. It would be interesting to test the $PM_{2.5}$-UHII correlation again in future studies of Beijing's UHI using rural air temperature measurements from weather stations situated in cleaner air with low $PM_{2.5}$ concentrations.

---

## Author Response (AR2)

1. **'I have a major concern regarding the use of only one LST image for model evaluation. Former model evaluations that used LST tried to acquire as many images as possible for one period. The argument that Landsat 8 images is more representative of the heterogeneity of the surface urban climate is receivable but not sufficient. These heterogeneities are also captured by MODIS data for instance, although smoothed by the lower resolution. Bechtel et al. 2019 who did a somparative study of LST in different cities using LCZ, MODIS and Landsat data found out that there was a great spatial correlation between the two. I believe they should be used jointly. Hu et al. (2014), Wouters et al. (2016) and Brousse et al. (2020) also used MODIS LST for evaluation of their model simulations. I would therefore like the authors to include such additional evaluation before acceptance of the manuscript.**

We understand the concerns regarding the limitations of using only one satellite image for this study. As explained in the manuscript, no further high resolution Landsat 8 images were available during the summer campaign period. Given the timescales involved in acquiring and analysing LSTs from alternative sources such as MODIS, and that I have now completed my PhD, this would be outside the scope of feasibility for this study. Furthermore, the information lost by smoothing the high resolution model output for comparisons with coarse resolution satellite data would considerably detract from the novelty of this particular study; an investigation of Beijing's neighbourhood-scale UHI characteristics. Additionally, as explained throughout the manuscript, given the inherent differences between LSTs and air temperatures, we are not expecting a high correlation in their spatial variabilities. The comparison is rather merely used as a guide and to highlight both the successful use of LCZ and OSM data to capture fine cool spaces and to help understand the air temperature-LST differences through the sensitivity studies. Given all of these reasons, we will not be adding further to this section of the manuscript.

2. **I still did not understand the rationale with having air temperature measured at the Pinggu station and upwind from the airport. The reasoning with the wind measurements being WMO standard makes sense. But then why not simply use the air temperature from the airport itself as it can be considered as rural too?**

We thank the editor for this comment. As explained in detail in the manuscript, rural air temperature measurements are perturbed in the model due to spatially varying urban land use characteristics and anthropogenic heat emissions. These rural measurements are ideally recorded at a site located upwind of the modelled urban area so that the air temperature sensor is not strongly influenced by the advection of urban heat; the urban heat island phenomenon is best expressed under stable conditions that inhibit the mixing of urban and rural air. The airport is located within urban Beijing (within the Sixth Ring Road); thus, the local environment is likely to be heavily affected by the surrounding urban surface thermal properties and nearby AHEs. Therefore, the use of air temperatures measured at the airport was not considered appropriate to drive the model. To clarify this, the manuscript is updated on Page 5 Line 197:

*"…an upwind measurement is desirable so that the air temperature sensor is not strongly influenced by the advection of urban heat. Air temperatures measured at the airport site were not deemed appropriate for this study given the local environment is likely heavily affected by the warming effects of surrounding urban surface thermal properties and nearby AHEs"*

3. **The implications on the results from adapting the surface resistance to evaporation for LCZ 4, 6 and 9 need to be discussed.**

We thank the editor for this comment. We lower the surface resistance to evaporation values of LCZ classes 4-6 and 9, relative to the 'fully' urban LCZ classes, as they are described by Stewart and Oke (2012) as consisting of significant pervious land cover. Assigning a surface resistance to evaporation value of 200 s m$^{-1}$ to all urban LCZ classes would lack physical sense. In Sect. 3.2 (Lines 455-465) we describe how the spatial heterogeneity in urban surface resistance to evaporation values produces greater spatial variability in modelled urban near-surface air temperatures and thus results in a stronger correlation with the spatial variability of LSTs. The high urban moisture case reduces the spatial correlation between modelled air temperatures and LSTs but increases the measured-modelled air temperature agreement at the IAP site. As explained in detail in the manuscript, this likely reflects the inherent differences between LSTs and air temperatures. It is unclear what further discussion the editor is requesting here.